# ZipperQuant: Bit-Based Inlier–Outlier Disaggregation for 4-Bit LLMs on GPU–CPU

## Abstract

Quantization has become a key technique for reducing the memory footprint and accelerating the inference of large language models (LLMs), especially as modern GPUs provide native INT4 compute units. However, quantizing both weights and activations to low bit-width often causes substantial accuracy degradation, since the limited representational range cannot simultaneously capture common values (*inliers*) and rare large-magnitude values (*outliers*). To address this challenge, we propose **ZipperQuant**, a novel 4-bit quantization paradigm for LLMs that disaggregates the computation of inliers and outliers across GPU and CPU. A key limitation of naive value-based disaggregation—offloading entire outlier values to the CPU—is that it suffers from the large performance gap between GPUs and CPUs and the high overhead of inter-device data transfer. ZipperQuant instead introduces a bit-based disaggregation strategy. Using smoothing, activation outliers are first absorbed into the weights, which are then split into low-order and high-order components. The GPU executes all inliers together with the low-order bits of outliers in low precision, while only the sparse high-order bits are offloaded to the CPU and multiplied with activations at high precision. These high-order CPU computations are further accelerated by a specialized lookup-table mechanism: since only a small set of bit patterns occurs, their results can be precomputed and reused, replacing costly multiplications with lightweight table lookups and accumulation, while also eliminating dequantization overhead. Extensive experiments demonstrate that ZipperQuant preserves near-FP16 accuracy while achieving up to $3.01\times$ speedup over a W4A16 baseline on an RTX 4090 with INT4 precision.

## 1 Introduction

Large language models (LLMs) have transformed the landscape of AI, redefining the boundaries of natural language processing and beyond (Grattafiori et al., 2024; Yang et al., 2025; Achiam et al., 2023; Team et al., 2023), and are now integral to applications ranging from everyday conversational assistants to high-stakes domains such as medical decision support (Thirunavukarasu et al., 2023) and code generation (Barke et al., 2023). This remarkable progress has been driven largely by scaling model size: LLMs have grown from 7B to 70B, and frontier models with over 600B parameters. However, this scaling results in a proportional increase in FLOPs and memory footprint, leading to a significant increase in matrix multiplications. These matrix multiplications are the primary cause of memory consumption and latency, which are major bottlenecks to practical deployment.

To address these efficiency bottlenecks, hardware vendors are increasingly adopting low-precision computation. For example, NVIDIA's Hopper and Blackwell architectures (NVIDIA, 2024) natively support both low-bit integer (INT8, INT4) and floating-point (FP8, FP4) formats, achieving significant throughput gains over traditional FP16 precision. Consequently, quantization, which compresses a model's weights and runtime activations into these low-bit representations to save memory and speed up inference (Ashkboos et al., 2025; Lin et al., 2025; Zhao et al., 2024b) has become a natural and indispensable strategy for scaling LLM deployment.

However, quantizing both weights and activations to low bit-width often leads to substantial accuracy degradation, as the limited representational range cannot simultaneously capture the dense distribution of common values (inliers) and the rare but large-magnitude values (outliers) (Dettmers et al., 2022). Early approaches such as SmoothQuant (Xiao, Guangxuan and Lin, Ji and Seznec,

Figure 1: Overview of ZipperQuant workflow.

Mickael and Wu, Hao and Demouth, Julien and Han, Song, 2023a) mitigate this issue by shifting activation outliers into the weights via channel-wise scaling. This works well at INT8, where static weights are relatively tolerant to a few additional outliers, but under more aggressive 4-bit quantization, weights already contain significant outliers, so further migrating activation outliers only amplifies quantization error. More recent methods adopt rotation-based techniques (Ashkboos et al., 2025; Liu et al., 2025; Sun et al., 2025), applying orthogonal transformations jointly to weights and activations to distribute outliers more evenly across channels. Yet, because activation outliers vary dynamically with input and context, most existing quantization approaches cannot fully eliminate them, leaving residual outliers that continue to degrade accuracy under aggressive quantization.

To address this challenge, we propose **ZipperQuant**, a novel 4-bit quantization paradigm for LLMs that disaggregates the computation of inliers and outliers across GPU and CPU. This separation directly tackles the difficulty of jointly quantizing weights and activations at low bit-width: inliers, which dominate the distribution, are preserved with high fidelity and executed efficiently on GPUs, while outliers, though rare but large in magnitude, are isolated and delegated to CPUs for high-precision handling. By preventing these extremes from distorting the overall quantization scale, ZipperQuant mitigates the accuracy loss that limits existing approaches.

A naive strategy is to offload entire outlier values to the CPU. However, this *value-based disaggregation* quickly becomes impractical: although outliers constitute only a small fraction of values, executing them in full on the CPU is highly inefficient. Modern GPUs provide hundreds of TOPS of INT4/INT8 throughput, whereas CPUs typically achieve only a few TOPS, resulting in a 10–100× performance gap. In addition, transferring entire outlier values over PCIe (32–64 GB/s) is more than 20× slower than accessing GPU HBM (> 1 TB/s). Together, these factors make value-based disaggregation dominated by CPU computation and inter-device data transfer overhead.

As shown in Figure 1, the central insight of ZipperQuant is that outliers need not be offloaded in full. Instead, weights are decomposed by bit significance: the dense low-order bits—including those of outliers—are executed on GPUs in low precision together with inliers, while only the sparse high-order bits are delegated to CPUs, where they are multiplied with activations at high precision. Limiting CPU execution to this high-order component fundamentally reduces both data transfer and computation overhead. To further accelerate CPU execution, we introduce a specialized lookup-table mechanism: because the number of possible high-order bit patterns is small, their products with activations can be precomputed and cached, enabling the CPU to replace costly multiplications with lightweight table lookups and accumulation. Finally, we co-design the inference engine to store high-order branches in sparse format and pipeline data transfer with computation, ensuring that CPU offloading does not become a bottleneck.

We summarize our contributions as follows:

- We propose **ZipperQuant**, a 4-bit quantization paradigm for LLMs that disaggregates the computation of inliers and outliers across GPU and CPU, directly addressing the accuracy loss of prior quantization approaches without efficiency degradation.
- We introduce a **bit-based disaggregation strategy** that executes inliers and the low-order bits of outliers on GPUs, while offloading only the sparse high-order bits to CPUs. This design reduces both computation and inter-device transfer, and further accelerates high-precision CPU execution with a lookup-table mechanism that replaces multiplications with lightweight lookups.
- We conduct extensive experiments on mainstream LLMs and system-level evaluations on both workstation (RTX 4090) and server-class (L20) GPUs, demonstrating that ZipperQuant preserves near-FP16 accuracy while achieving up to 3.01× speedup over a W4A16 baseline on RTX 4090

with INT4 precision, and matches or surpasses state-of-the-art W4A4/W4A8 systems in end-to-end throughput under comparable accuracy.

## 2 RELATED WORK

**Post-Training Quantization.** Post-training quantization is a widely used technique for reducing Transformer model size and accelerating inference (Lin et al., 2025). Weight-only quantization (Lin et al., 2024; Frantar et al., 2023) compresses weights to low bit-widths but typically upcasts them back to full precision during inference. To further improve efficiency, recent work investigates joint quantization of both weights and activations for low-bit inference (Li et al., 2025; Zhao et al., 2025). SmoothQuant (Xiao, Guangxuan and Lin, Ji and Seznec, Mickael and Wu, Hao and Demouth, Julien and Han, Song, 2023b) first demonstrates effective W8A8 quantization, while subsequent methods achieve 4-bit precision using learnable quantization parameters (Shao et al., 2024) or orthogonal rotations (Liu et al., 2025; Ashkboos et al., 2025), improving robustness against outliers and better exploiting low-precision hardware units. Our approach complements these efforts by introducing outlier-aware optimization for 4-bit weight–activation quantization, while remaining orthogonal to learnable quantization and rotation-based methods. In addition, several studies assign different bit-widths to different components of activations and weights—for example, computing outliers in higher precision (Zhao et al., 2024b; Huang et al., 2025). Unlike most of these mixed-precision approaches, which focus on value-level disaggregation and operate solely on GPUs, ZipperQuant explores finer-grained bit-level strategies with heterogeneous GPU–CPU co-execution, achieving both efficiency and accuracy.

**Hybrid GPU–CPU Inference.** Hybrid execution has been explored to alleviate the memory footprint of LLMs, which arises from both model parameters and intermediate activations and often exceeds the capacity of commodity GPUs. Prior work can be broadly divided into *parameter and KV-cache offloading* (Wang et al., 2025; Sheng et al., 2023; Fang et al., 2025) and *hybrid computation* (Zhao et al., 2024a; Song et al., 2024; Cao et al., 2025; ggml org, 2025; Xu et al., 2025). The former treats the CPU primarily as a memory reservoir, dynamically transferring parameters to the GPU for execution. The latter leverages the CPU as an auxiliary compute device to offload complementary tasks—such as layers that exceed GPU memory capacity (ggml org, 2025), attention operations involving KV caches stored in CPU memory (Cao et al., 2025; Sheng et al., 2023), neurons with low activation frequency (Song et al., 2024), or activation outlier channels (Xu et al., 2025)—in parallel with GPU execution. Our study also falls into the second category but specifically targets quantized inference. Whereas existing approaches offload relatively coarse-grained units—such as layers, tensors, channels, or neurons—to the CPU, we introduce a much finer-grained strategy that disaggregates computation at the bit level, executing inliers and the low-order bits of outliers on GPUs while delegating only the sparse high-order bits to CPUs.

## 3 QUANTIZATION PRELIMINARY

Quantization is a widely used technique to accelerate the linear operators in Transformer layers by reducing their arithmetic precision. For a tensor $\mathbf{X}$, uniform symmetric quantization with $k$ bits is defined as

$$\mathbf{Q}_k^\mathbf{X} = \text{round}\left(\frac{\mathbf{X}}{s_k^\mathbf{X}}\right), \quad s_k^\mathbf{X} = \frac{\max(|\mathbf{X}|)}{q_{\max}}, \tag{1}$$

where $\mathbf{Q}_k^\mathbf{X}$ is the $k$-bit integer representation of $\mathbf{X}$, $s_k^\mathbf{X}$ is the scaling factor, and $q_{\max}$ is the maximum representable integer. For signed integers with $k$ bits, $q_{\max} = 2^{k-1} - 1$. The dequantized tensor is then given by $Q_k(\mathbf{X}) = s_k^\mathbf{X} \mathbf{Q}_k^\mathbf{X}$. For a linear layer with activation $\mathbf{X}$ and weight $\mathbf{W}$,

$$\mathbf{X}\mathbf{W} \approx Q_k(\mathbf{X}) Q_k(\mathbf{W}) = s_k^\mathbf{X} s_k^\mathbf{W} \cdot \mathbf{Q}_k^\mathbf{X} \mathbf{Q}_k^\mathbf{W}. \tag{2}$$

To exploit low-precision compute units on modern GPUs, $\mathbf{Q}_k(\mathbf{X})$ and $\mathbf{Q}_k(\mathbf{W})$ typically adopt the same bit-width. Otherwise, one operand must be dequantized at runtime to match the other, which undermines the performance advantage of quantized execution. Following prior work (Lin et al., 2025; Li et al., 2025), we denote $x$-bit weights and $y$-bit activations as $\mathbf{W}x\mathbf{A}y$ (such as W4A4 denotes 4-bit weights and 4-bit activations) and focus on signed integer quantization. The formulation for unsigned integers is deferred to Appendix C.1.

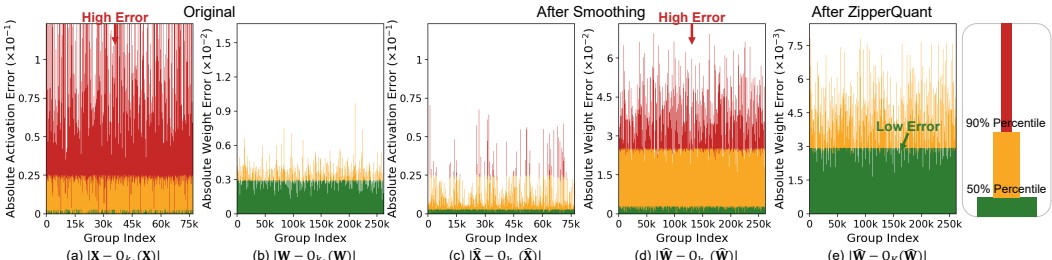

Figure 2: Quantization error visualization on LLaMA3-8B. Red denotes large errors, while green denotes small ones. (a,b) High activation and weight errors in the model. (c,d) After smoothing, activation errors are largely shifted into weights. (e) ZipperQuant substantially reduces weight errors.

## 4 METHOD

In this section, we first analyze the sources of quantization error and derive their theoretical upper bounds. We then introduce **ZipperQuant**, which "zips" low-order and high-order bits across GPU and CPU to combine the throughput of low-precision execution with the accuracy of higher-bit processing. Finally, we present a co-designed Zipper engine that enables efficient parallel co-execution, yielding tighter error bounds and a more favorable efficiency–accuracy trade-off.

### 4.1 PROBLEM FORMULATION

Consider a Transformer linear layer (Vaswani et al., 2017) with activation $\mathbf{X} \in \mathbb{R}^{t \times m}$ and weight $\mathbf{W} \in \mathbb{R}^{m \times n}$, where $t$ is the token length and $m, n$ are the input and output channel dimensions. The $k$-bit quantization error is defined as

$$E(\mathbf{X}, \mathbf{W}) = \|\mathbf{X}\mathbf{W} - Q_k(\mathbf{X})Q_k(\mathbf{W})\|_F, \tag{3}$$

where $\|\cdot\|_F$ is the Frobenius norm.

**Theorem 4.1** (Upper bound for quantized linear layer error). *The quantization error of each linear layer admits the following upper bound:*

$$E(\mathbf{X}, \mathbf{W}) \le \|\mathbf{X}\|_F \|\mathbf{W} - Q_k(\mathbf{W})\|_F + \|Q_k(\mathbf{W})\|_F \|\mathbf{X} - Q_k(\mathbf{X})\|_F. \tag{4}$$

The proof is deferred to Appendix B.1. This bound reveals that the error depends on four terms: the magnitudes of the activation and the quantized weight, $\|\mathbf{X}\|_F$ and $\|Q_k(\mathbf{W})\|_F$, and the quantization errors of weight and activation, $\|\mathbf{W} - Q_k(\mathbf{W})\|_F$ and $\|\mathbf{X} - Q_k(\mathbf{X})\|_F$. Minimizing the overall quantization error thus requires jointly controlling all four terms.

### 4.2 ZIPPERQUANT: ZIP-THE-BYTE QUANTIZATION FOR HYBRID EXECUTION

**Migrating outliers from activations to weights.** Smoothing (Xiao, Guangxuan and Lin, Ji and Seznec, Mickael and Wu, Hao and Demouth, Julien and Han, Song, 2023a) is a common approach to reduce activation magnitudes. With per-channel scaling factors $\boldsymbol{\lambda} \in \mathbb{R}^m$, outliers can be shifted from activations into weights. As shown in Figure 2(c), the smoothed activation $\hat{\mathbf{X}} = \mathbf{X} \operatorname{diag}(\boldsymbol{\lambda})^{-1}$ has lower magnitudes and fewer outliers, reducing activation quantization error. However, as in Figure 2(d), the transformed weights $\hat{\mathbf{W}} = \mathbf{W} \operatorname{diag}(\boldsymbol{\lambda})$ exhibit larger magnitudes and more outliers, thereby increasing weight quantization error. Consequently, the terms $\|\mathbf{X}\|_F$ and $\|\mathbf{X} - Q_k(\mathbf{X})\|_F$ in Equation 4 decrease, while our next objective is to optimize the remaining two terms: the magnitude of $Q_k(\mathbf{W})$ and the weight quantization error $\|\mathbf{W} - Q_k(\mathbf{W})\|_F$.

**Theorem 4.2** (Upper bound for quantized weight error). *The quantized weight error satisfies*

$$\|\mathbf{W} - Q_k(\mathbf{W})\|_F \le \frac{\max(|\mathbf{W}|)\sqrt{mn}}{2q_{\max}}. \tag{5}$$

See Appendix B.2 for the proof. The error depends on three factors: the maximum absolute weight value $\max|\mathbf{W}|$, the matrix size $\sqrt{mn}$, and the maximum representable integer $q_{\max}$. The first two are determined by the weight distribution and layer dimensionality. The most direct way to reduce

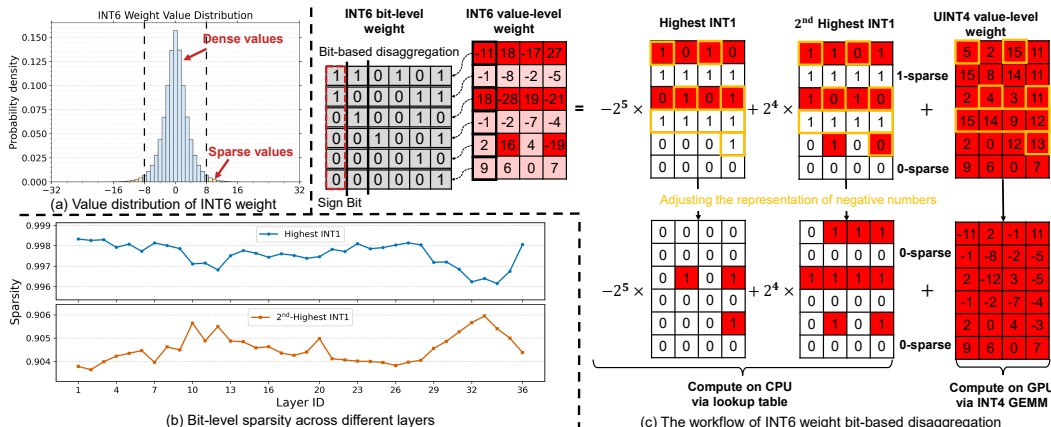

Figure 3: (a) Distribution of INT6 quantized $K$ projection weight in layer 1 of Qwen3 4B. (b) Bit-level sparsity of the $K$ projection weights across layers in Qwen3 4B. (c) Workflow of INT6 weight bit-based disaggregation: decompose INT6 into the highest one bit, the second highest one bit, and a UINT4 remainder, we evaluate the first two on the CPU via lookup tables, and compute the UINT4 component on the GPU using GEMM.

the error is to increase $q_{\max}$. However, directly enlarging $q_{\max}$ comes at two costs: (i) larger memory footprint and bandwidth consumption, since storage and traffic scale roughly linearly with bit-width; and (ii) reduced compute intensity, because current GPU low-precision units are optimized for 4- and 8-bit arithmetic, so unsupported bit-widths require runtime packing and unpacking.

Motivated by this, our goal is to design a new quantization paradigm that *effectively expands the representational range of weights—i.e., indirectly increases $q_{\max}$—without incurring the hidden costs of higher bit-widths*. To illustrate this point, we first consider setting a relatively high $q_{\max}$, corresponding to INT6 quantization. As shown in Figure 3(a), most weight values then fall within a narrow low-magnitude interval (e.g., $[-8, 8]$), while only a few sparse outliers extend to larger magnitudes. This naturally suggests offloading those outlier computations to the CPU, where they can be processed in high precision, while keeping inliers on the GPU in low precision. Such a design ensures that $q_{\max}$ is still represented under high precision, while the bulk of GPU computation and memory access remains low-bit. However, value-based offloading requires transferring entire outlier values to the CPU and computing them in full, which inflates both inter-device traffic and CPU workload, making the approach highly inefficient.

**Bit-based disaggregation for weight outliers with hybrid execution.** To address the limitations of value-based offloading, we introduce a bit-based disaggregation strategy that assigns the high-order bits of outliers to the CPU while retaining the low-order bits on the GPU for parallel execution. In this way, high bit-width accuracy is preserved for the smoothed weights, while the GPU continues to benefit from low-precision parallel acceleration.

As shown in Figure 3(b), outliers at the high-order bit level exhibit high sparsity across different layers. We therefore partition the $K$-bit representation: the lower $k_1$ bits are executed on the GPU, matched to its native low-precision compute units, while the higher $k_2$ bits are delegated to the CPU. See Appendix E.5 for more details. This design leverages the throughput advantage of low-precision GPU computation while exploiting the sparsity of high-order bits on the CPU, thereby maintaining accuracy. The quantization with bit-based disaggregation in ZipperQuant can be expressed as:

$$Q_K(\hat{\mathbf{W}}) = s_K^{\mathbf{W}} \mathbf{Q}_K^{\mathbf{W}} = s_K^{\mathbf{W}} \left( \mathbf{U}_{k_1}^{\mathbf{W}} + 2^{k_1} \cdot \mathbf{Q}_{k_2}^{\mathbf{W}} \right) = U_{k_1}(\hat{\mathbf{W}}) + 2^{k_1} Q_{k_2}(\hat{\mathbf{W}}), \tag{6}$$

where $\mathbf{U}_{k_1}^{\mathbf{W}}$ and $\mathbf{Q}_{k_2}^{\mathbf{W}}$ denote the $k_1$-bit unsigned and $k_2$-bit signed representations of $\mathbf{W}$, respectively. The proof of this decomposition is provided in Appendix B.3. Accordingly, the matrix multiplication $\mathbf{X}\mathbf{W}$ can be approximated as:

$$\mathbf{X}\mathbf{W} = \hat{\mathbf{X}}\hat{\mathbf{W}} \approx \underbrace{Q_{k_1}(\hat{\mathbf{X}})\,U_{k_1}(\hat{\mathbf{W}})}_{\text{GPU computation}} + \underbrace{\hat{\mathbf{X}} \cdot 2^{k_1} Q_{k_2}(\hat{\mathbf{W}})}_{\text{CPU computation}}. \tag{7}$$

Figure 4: (a) Naive CPU execution of high-order bits leads to excessive latency. (b) Workflow of our LUT-based design: only weight rows containing outliers are stored, corresponding activations are selectively transferred, grouped, and used to index precomputed LUT entries, which are accumulated to form partial sums before being merged with GPU results.

The first term corresponds to low-order bits executed on the GPU using INT4 GEMM, while the second term corresponds to high-order bits executed on the CPU in FP16. These two terms are computed in parallel, and implementation details are provided in Section 4.3.

**Error bound analysis.** Based on Equation 7, we can derive the upper bound of ZipperQuant error:

**Theorem 4.3** (Upper bound for ZipperQuant error). *The upper bound of ZipperQuant's quantization error is given by*

$$E(\hat{\mathbf{X}}, \hat{\mathbf{W}}) \leq \|\hat{\mathbf{X}}\|_F \|\hat{\mathbf{W}} - U_{k_1}(\hat{\mathbf{W}}) - 2^{k_1}Q_{k_2}(\hat{\mathbf{W}})\|_F + \|U_{k_1}(\hat{\mathbf{W}})\|_F \|\hat{\mathbf{X}} - Q_{k_1}(\hat{\mathbf{X}})\|_F. \quad (8)$$

The proof is provided in Appendix B.4. Compared with Theorem 4.1, the upper bound obtained here is tighter. For the weight quantization error term $\|\mathbf{W} - Q_{k_1}(\mathbf{W})\|_F$, ZipperQuant benefits from an effectively larger $q_{\max}$, as discussed in Theorem 4.2. For the magnitude of $Q_{k_1}(\mathbf{W})$, note that it depends on the scaling factor $s_{k_1}^{\mathbf{W}}$. According to Equation 1, a larger bit-width yields a smaller scaling factor. In conventional quantization, the scale is determined by the lower bit $k_1$, whereas in our design it is determined by the higher bit $K$, which further tightens the error bound.

## 4.3 ZIPPER ENGINE FOR GPU–CPU CO-EXECUTION

**Bit-level LUT for CPU acceleration.** Although the high-order bits are extremely sparse and contribute little arithmetic cost in theory, executing their matrix multiplications on the CPU still incurs substantial latency. As shown in Figure 4(a), the CPU path is nearly $100\times$ slower than the 4-bit GPU path, preventing fully parallel execution. The main bottleneck is that, despite targeting sparse low-bit arithmetic, CPUs lack native low-bit support and must execute the high-order branch in widened precision after dequantization, which negates the low-bit advantage. To overcome this, we introduce a *bit-level lookup table (LUT) mechanism* that transforms multiplications with sparse high-order bits into indexed table lookups, thereby eliminating dequantization and reducing overhead. Moreover, since weights are much larger than activations and remain fixed during inference, precomputing their high-order contributions enables frequent reuse. This makes lookup-based computation substantially more efficient than direct multiplications, which repeatedly process the same weight patterns with limited reuse. While inspired by the observation in T-MAC (Wei et al., 2025) that LUT-based computation can be efficient on CPUs, our approach fundamentally differs: rather than building a CPU-only engine, we integrate LUTs into a heterogeneous GPU–CPU workflow and tailor them specifically for the sparse high-order bits in our bit-based disaggregation framework.

Specifically, as illustrated in Figure 4(b), we retain offline only the weight rows containing outliers (e.g., indices 0, 2, and 4). During inference, we transfer to the CPU only the activations at the corresponding column positions. These activations are then grouped along the column dimension with group size $g$ (e.g., $g = 3$). For each group, we precompute and cache all $2^g - 1$ nonzero linear combinations in a LUT, which can be repeatedly reused across multiple weight rows. This procedure replaces GEMM with indexed LUT lookup followed by accumulation. Furthermore, the fine-grained grouping strategy reveals intra-group sparsity: if a group's bit pattern is '000' (e.g., columns 1 and 3), its contribution is null and can be skipped. Otherwise, the corresponding precomputed value is fetched by index and accumulated into the partial sum. Finally, partial sums from all groups are aggregated on the CPU and returned to the GPU in a single DtoH transfer for integration.

Table 1: Comparison of the perplexity score on WikiText2 and averaged accuracy on six zero-shot commonsense reasoning tasks. The results of LLaMA3 for SmoothQuant Xiao, Guangxuan and Lin, Ji and Seznec, Mickael and Wu, Hao and Demouth, Julien and Han, Song (2023b), GPTQ (Frantar et al., 2023), QuaRot (Ashkboos et al., 2025) were obtained using their publicly released codebase, while AWQ (Lin et al., 2024), SpinQuant (Liu et al., 2025) results were quoted from their papers. Full results are in the Appendix E.2.

| Precision | Method | LLaMA-3.2 1B | | LLaMA-3.2 3B | | LLaMA3 8B | | Qwen3 4B | | Qwen3 8B | | Qwen3 14B | | Qwen3 32B | |
|---|---|---|---|---|---|---|---|---|---|---|---|---|---|---|---|
| | | 0-shot Avg. | Wiki (↓) | 0-shot Avg. | Wiki (↓) | 0-shot Avg. | Wiki (↓) | 0-shot Avg. | Wiki (↓) | 0-shot Avg. | Wiki (↓) | 0-shot Avg. | Wiki (↓) | 0-shot Avg. | Wiki (↓) |
| W16A16 | – | 60.1 | 13.4 | 66.2 | 10.7 | 74.2 | 6.1 | 63.9 | 13.7 | 67.4 | 9.7 | 70.9 | 8.6 | 71.8 | 7.6 |
| W4A16 | RTN | 58.2 | 20.7 | 60.9 | 18.8 | 69.7 | 8.2 | 60.2 | 17.6 | 64.4 | 12.0 | 67.0 | 9.9 | 61.7 | 38.5 |
| | GPTQ | 57.6 | 17.3 | 61.7 | 15.2 | 68.6 | 7.2 | 61.4 | 14.5 | 66.0 | 10.3 | 69.5 | 9.2 | 70.0 | 8.3 |
| | AWQ | 57.9 | 15.6 | 63.0 | 12.7 | 71.2 | 7.3 | 61.4 | 16.6 | 69.4 | 10.5 | 65.9 | 9.6 | 70.5 | 8.2 |
| W4A8 | RTN | 56.5 | 20.7 | 60.7 | 29.0 | 67.9 | 8.2 | 55.8 | 30.6 | 60.8 | 12.3 | 65.3 | 10.9 | 64.4 | 11.2 |
| | SmoothQuant | 49.1 | 108.2 | 59.8 | 288.5 | 64.4 | 10.7 | 57.0 | 22.6 | 60.8 | 12.5 | 65.0 | 11.0 | 64.1 | 18.2 |
| | QuaRot | 55.7 | 18.9 | 62.3 | 12.4 | 65.2 | 7.8 | 57.8 | 16.7 | 63.2 | 12.9 | 67.6 | 10.4 | 67.9 | 10.5 |
| | SpinQuant | 58.2 | 15.3 | 63.8 | 11.6 | 72.1 | 6.7 | 57.6 | 17.1 | 63.9 | 12.1 | 67.9 | 10.2 | 68.2 | 10.3 |
| | ZipperQuant | 58.9 | 15.1 | 64.3 | 11.1 | 72.5 | 6.6 | 60.1 | 15.3 | 64.6 | 11.5 | 68.3 | 9.9 | 69.1 | 9.5 |
| W4A4 | RTN | 42.6 | 137.5 | 43.2 | 741.9 | 45.4 | 241.6 | 38.3 | 8791 | 37.2 | 4392 | 38.4 | 18749 | 40.3 | 1796 |
| | SmoothQuant | 38.9 | 2027.5 | 44.7 | 372.3 | 37.9 | 867.5 | 37.3 | 9910 | 37.6 | 3360 | 37.5 | 21675 | 40.2 | 1806 |
| | QuaRot | 48.9 | 50.3 | 53.4 | 26.9 | 61.5 | 21.4 | 53.2 | 21.5 | 60.1 | 24.5 | 64.1 | 18.2 | 64.6 | 15.6 |
| | SpinQuant | 52.0 | 44.8 | 57.5 | 22.4 | 65.5 | 18.6 | 54.2 | 21.7 | 60.6 | 25.9 | 64.6 | 17.5 | 64.9 | 16.0 |
| | ZipperQuant | 54.2 | 28.9 | 60.6 | 20.6 | 67.6 | 16.5 | 57.0 | 18.9 | 62.0 | 21.4 | 66.0 | 12.0 | 66.1 | 12.4 |

**Unifying bit-plane sparsity via sign restoration.** Although computation can be reduced by evaluating only the channels containing weight outliers on the CPU, naively discarding the remaining channels introduces substantial error. A key challenge is that high-order weight bits often encode sign information that is not captured by the GPU computation. Moreover, under two's-complement encoding, negative values appear "1-sparse" in the high bits, while positive values are "0-sparse", resulting in asymmetric channel statistics. To address this, we equivalently rewrite the terms in Equation 7. As shown in Figure 3(c), when weights are stored as UINT4 on the GPU, nonnegative entries remain unchanged. For negative entries, we perform *sign restoration*: subtracting 16 at those positions on the GPU, while adding 16 to the corresponding CPU binary representation. This procedure both unifies the sparsity pattern and ensures correct handling of negative values. In practice, we further divide GPU weights by 2 and compensate by multiplying the quantization scale by 2, thereby preventing overflow in the INT4 representation.

## 5 EXPERIMENTS

### 5.1 SETUPS

**Models and metric.** We benchmark against two mainstream LLMs: the LLaMA-3 (Grattafiori et al., 2024) models (1B/3B/8B) and Qwen3 (Yang et al., 2025) models (4B/8B/14B/32B). Following previous works (Sun et al., 2025; Liu et al., 2025), we random samples the 128 prompts from C4 dataset (Raffel et al., 2020) for calibration, each sentence with 2028 tokens. To evaluate the commonsense reasoning capability of our method, we use six zero-shot evaluation tasks, including ARC-Challenge, ARC-Easy (Clark et al., 2018), HellaSwag (Zellers et al., 2019), LAMBADA (Paperno et al., 2016), PIQA(Bisk et al., 2020), and WinoGrande (Sakaguchi et al., 2021). Additionally, we also report the perplexity score on WikiText2 (Merity et al., 2017) datasets. All accuracy metrics are tested on lm-eval (Gao et al., 2024). We also evaluate the memory save and speedup.

**Baselines.** We compare ZipperQuant against six popular PTQ methods, including the naive quantization method RTN (Yao et al., 2023), the weight-only quantization GPTQ (Frantar et al., 2023) and AWQ (Lin et al., 2024), the weight-activation quantization SmoothQuant (Xiao, Guangxuan and Lin, Ji and Seznec, Mickael and Wu, Hao and Demouth, Julien and Han, Song, 2023b) and two recent state-of-the-art methods QuaRot (Ashkboos et al., 2025), SpinQuant (Liu et al., 2025). See Appendix D.1 for more details.

**Implementation details.** Please refer to Appendix E.1 fore more details.

**Hardware.** We evaluate ZipperQuant on two GPU–CPU platforms. *Workstation*: an NVIDIA RTX 4090 (24 GB) GPU paired with an Intel Xeon Gold 6430 CPU and 120 GB DDR5 host memory.

*Inference server*: an NVIDIA L20 (48 GB) GPU paired with an Intel Xeon Platinum 8457C CPU and 100 GB DDR5 host memory.

Unless otherwise specified, the main figures and tables in this section report results on the RTX 4090 workstation, and we provide additional throughput results on the L20 server in Appendix E.3.

## 5.2 ACCURACY RESULTS

Table 1 presents the comparison of WikiText-2 perplexity and average accuracy on six zero-shot tasks for LLaMA-3 and Qwen3 models. See more details in Appendix E.2.

**Language generation tasks.** In the W4A8 setting, ZipperQuant outperforms all baselines and achieves performance comparable to W16A16. For examples, on LLaMA3.2-3B, it improves over the rotation-based SpinQuant by 0.5 perplexity points and reduces the gap to full precision to 0.4. Moreover, ZipperQuant generally outperforms W4A16 across LLaMA models, lowering perplexity by 0.5 to 1.6 points, attributable to its bit-based disaggregation that preserves outlier information. When activations are further quantized to 4-bits, most methods exhibit substantial increases in perplexity (e.g., SpinQuant and QuaRot), and some fail to produce meaningful results (e.g., SmoothQuant and RTN). ZipperQuant closes this gap by up to 15 points.

**Zero-shot tasks.** In the 4-bit weight and 8-bit activation setting, ZipperQuant achieves accuracy comparable to W4A16 and in some cases surpasses it. For instance, on LLaMA-3.2 1B and 3B, it exceeds AWQ by 0.7% and 1.3%, respectively. Additionally, when comes to W4A4, ZipperQuant achieves strong performance without any retraining (as required by SpinQuant) and without incurring online conversion overhead (as in QuaRot and SpinQuant). On Qwen family models, it provides a 2.1% to 3.2% improvement, substantially outperforming prior state of the art. Nevertheless, since both weights and activations are quantized to 4-bits, a gap to full precision still remains. ZipperQuant respectively reduces this gap by up to 5.9% and 4.7% on LLaMA and Qwen models.

## 5.3 PERFORMANCE ANALYSIS

**End-to-end speedups.** Figure 5 summarizes the end-to-end gains of ZipperQuant over FP16 TensorRT-LLM and recent quantization systems. On an RTX 4090, ZipperQuant (W4A4) reaches 124.8–817.9 tokens/s across batch sizes 1–8, delivering about $1.8$–$2.1\times$ higher decoding throughput than the FP16 TensorRT-LLM engine and $1.6$–$2.3\times$ over AWQ. Relative to prior W4A4 methods, it is $7$–$14\times$ faster, while staying within roughly $3$–$22\%$ of the W4A8 QServe system despite using lower-precision weights. Because both weights and activations are quantized and kept on GPU, ZipperQuant accelerates both the compute-bound prefill phase (up to $3.01\times$) and the memory/IO-bound decoding phase (up to $1.90\times$), and enables models such as Qwen3-14B to run fully in 24 GB GPU memory without CPU offloading, yielding up to $3.10\times$ end-to-end speedup.

**End-to-end memory save.** Figure 6 shows the GPU memory footprint during inference for LLaMA3 8B and Qwen3 14B. ZipperQuant reduces the original 8B model size from 16.4 GB to 4.4 GiB, yielding overall compression ratios of $3.73\times$ and $3.93\times$ on 8B and 14B, respectively. Since both weights and activations are quantized, our method further reduces memory by an additional 0.4 GiB on both the 8B and 14B models relative to weight-only AWQ.

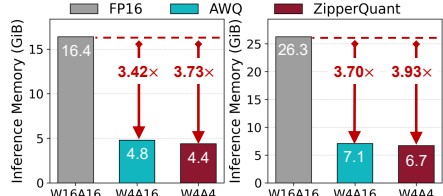

Figure 6: Inference Memory of ZipperQuant and baselines on LLaMA3 8B and Qwen3 14B.

**Cost of the CPU path.** A natural concern is that sending activations to the CPU for high-order bit LUT computations might introduce prohibitive overhead. However, our implementation pipelines activation quantization and device-to-host transfers with the GPU INT4 GEMM, so the two processes run in parallel rather than sequentially. The latency breakdown in Appendix Table 8 shows that CPU LUT computation and PCIe transfers contribute only a bounded additional cost and do not dominate the end-to-end latency, which is consistent with the strong throughput reported in Table 2.

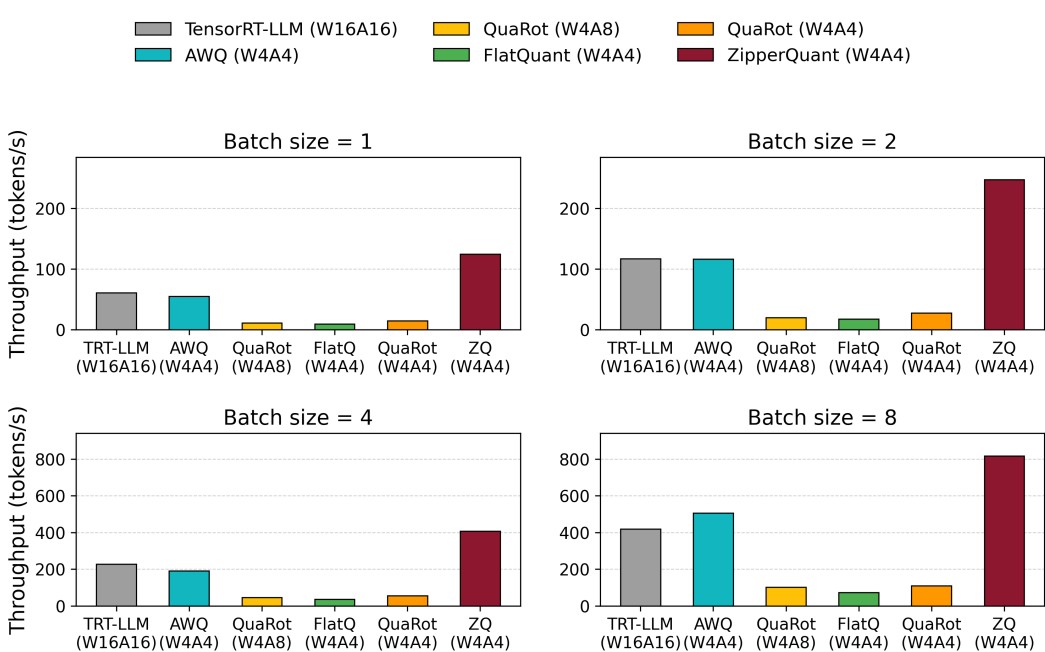

Figure 5: End-to-end throughput and speedups of ZipperQuant (W4A4) compared to FP16 TensorRT-LLM, AWQ, QuaRot, FlatQuant, and QServe on an RTX 4090.

Table 2: End-to-end decoding throughput (tokens/s) under different batch sizes. All methods are evaluated on the same model and GPU with greedy decoding.

| Method | Batch size=1 | Batch size=2 | Batch size=4 | Batch size=8 |
|---|---|---|---|---|
| TensorRT-LLM (W16A16) | 61.18 | 117.10 | 228.00 | 418.20 |
| AWQ (W4A4) | 55.20 | 116.50 | 191.80 | 506.40 |
| QuaRot (W4A8) | 11.20 | 20.40 | 46.20 | 101.20 |
| QServe (W4A8) | 128.90 | 264.24 | 520.70 | 1015.20 |
| FlatQuant (W4A4) | 9.40 | 17.90 | 35.72 | 72.66 |
| QuaRot (W4A4) | 14.90 | 27.70 | 55.30 | 110.30 |
| ZipperQuant (W4A4) | 124.80 | 247.20 | 407.80 | 817.90 |

## 5.4 INTEGRATE WITH SPINQUANT

Rotation-based quantization reduces outlier magnitudes through online and offline rotations, but direct 4-bit quantization still introduces substantial quantization error. Zipper engine mitigates this by employing bit-based disaggregation to offload high-order outlier bits to the CPU, thereby preserving GPU compute efficiency and inference accuracy. Table 3 presents results that based on SpinQuant on LLaMA3 3B and 8B, the Zipper engine disaggregated INT6 into INT2 on the CPU and INT4 on the GPU, and evaluates six zero-shot tasks along with WikiText-2 perplexity. Zipper lowers Spin-Quant perplexity by 6.9 and 6.3 points on the 3B and 8B models, respectively. Moreover, bit-based disaggregation improves average accuracy by 3.2% for 3B and 1.5% for 8B on zero-shot tasks, particularly gains up to 4.4% on ARC-C task.

## 5.5 ABLATION STUDY

As illustrate in Table 4, we present a series of ablation studies for SpinQuant on the LLaMA family 3B and 8B models. First, under 4-bit quantization, both naive quantization and smoothing lead

Table 3: SpinQuant integrated with Zipper: INT6 disaggregated to 2-bits on CPU and 4-bits on GPU on LLaMA3 3B and 8B, reporting zero-shot accuracy on six tasks and WikiText-2 perplexity.

| Model | Precision | Method | ARC-C | ARC-E | HellaSwag | PIQA | Winogrande | LAMBADA | Avg. | Wiki2 ($\downarrow$) |
|---|---|---|---|---|---|---|---|---|---|---|
| LLaMA -3.2 3B | W16A16 | – | 47.6 | 69.9 | 71.0 | 76.0 | 66.6 | 65.9 | 66.2 | 10.7 |
| | W4A4 | SpinQuant | 43.7 | 54.6 | 56.3 | 66.7 | 65.5 | 57.9 | 57.5 | 22.4 |
| | | ZipperQuant | 45.8 | 58.1 | 63.5 | 70.3 | 64.5 | 61.6 | 60.6 | 20.6 |
| | | Spin+Zipper | 47.9 | 63.6 | 66.1 | 71.0 | 64.8 | 62.9 | 63.8 | 15.3 |
| LLaMA3 8B | W16A16 | – | 57.7 | 77.6 | 79.6 | 80.7 | 73.7 | 75.6 | 62.7 | 6.1 |
| | W4A4 | SpinQuant | 45.9 | 66.5 | 75.9 | 68.0 | 68.5 | 67.9 | 65.5 | 18.6 |
| | | ZipperQuant | 47.9 | 69.2 | 73.5 | 73.6 | 71.3 | 70.1 | 67.6 | 16.5 |
| | | Spin+Zipper | 52.3 | 69.4 | 73.0 | 74.9 | 71.5 | 73.7 | 69.1 | 12.9 |

Table 4: Ablation study of ZipperQuant on LLaMA-3.2 3B and LLaMA3 8B, reporting the average accuracy on six zero-shot tasks and WikiText-2 perplexity.

Figure 7: Ablation study of ZipperQuant prefill and decoding throughput on LLaMA3 8B.

| Precision | Method | LLaMA-3.2 3B | | LLaMA3 8B | |
|---|---|---|---|---|---|
| | | 0-shot Avg. | Wiki ($\downarrow$) | 0-shot Avg. | Wiki ($\downarrow$) |
| W16A16 | – | 66.2 | 10.7 | 74.2 | 6.1 |
| W4A4 | Naive 4-bits | 41.2 | 743.9 | 46.2 | 243.7 |
| | Smoothing | 44.6 | 372.3 | 37.8 | 867.5 |
| | w/o Smoothing | 46.5 | 265.8 | 45.1 | 437.5 |
| | ZipperQuant | 60.6 | 20.6 | 67.6 | 16.5 |

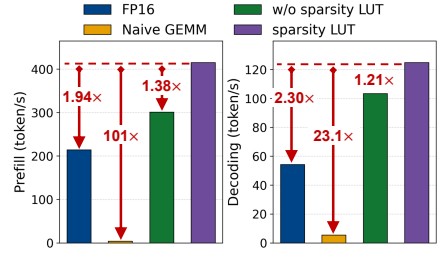

to substantial accuracy degradation. Although smoothing the quantized results offers a slight improvement over conventional quantization, the performance remains unsatisfactory. Compared with applying ZipperQuant directly without smoothing, zero-shot accuracy decreases by at least 15.1%, and perplexity increases markedly. While this preserves weight precision, it fails to account for outliers in activations which have larger magnitudes (see Figure 2), thereby causing a significant increase in activation quantization error.

Figure 7 compares the impact of different CPU computation strategies on prefill and decoding throughput for LLaMA3-8B. Executing high-order bits on the CPU with naive GEMM introduces substantial latency and can underperform the FP16 baseline. The slowdown is most pronounced in prefill, reaching up to $101\times$, due to online dequantization of high-order bits and the CPU's lower throughput for large matrix multiplications relative to the GPU. Replacing GEMM with a non-sparse LUT reduces latency but remains slower than the sparse LUT by $1.38\times$ in prefill and $1.21\times$ in decoding. Overall, decoding is less costly than prefill since it processes fewer activations, which shortens CPU online table construction.

## 6  CONCLUSION

In this work, we introduce a novel 4-bit quantization paradigm ZipperQuant that achieves accurate, efficient LLM inference via bit-based disaggregation across GPU and CPU. We further develop a Zipper engine that enables seamless parallel execution on GPU and CPU. On the CPU, matrix operations for high-order bit planes are replaced with LUT computations, which avoid dequantization overhead. On mainstream LLMs using an RTX 4090, ZipperQuant maintains accuracy and achieves up to a $3.01\times$ speedup over a W4A16 baseline, demonstrating practical W4A4 inference when outliers are handled on both GPU and CPU. This result supports efficient deployment of large-scale LLMs on heterogeneous GPU-CPU systems and broadens the scope of interactive AI applications.

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

## A    THE USE OF LLMS

We employed LLMs (e.g., ChatGPT) solely for editorial assistance (grammatical, phrasing, and stylistic refinement). This model did not participate in the generation of primary methods, analytical implementation, or experimental design, nor did it influence research methodology, results, or conclusions. All technical content was authored and verified by the authors.

## B    PROOFS

### B.1    PROOF OF THEOREM 1

**Theorem 4.1.** The $k$-bit quantization error $E(\mathbf{X}, \mathbf{W}) = ||\mathbf{X}\mathbf{W} - Q_k(\mathbf{X})Q_k(\mathbf{W})||_F$ upper bound can be computed as follows:

$$E(\mathbf{X}, \mathbf{W}) \leq ||\mathbf{X}||_F ||\mathbf{W} - Q_k(\mathbf{W})||_F + ||Q_k(\mathbf{W})||_F ||\mathbf{X} - Q_k(\mathbf{X})||_F. \quad (9)$$

*Proof.*

$$\begin{aligned}
\|\mathbf{XW} - Q_k(\mathbf{X})Q_k(\mathbf{W})\|_F &= \|\mathbf{XW} - \mathbf{X}Q_k(\mathbf{W}) + \mathbf{X}Q_k(\mathbf{W}) - Q_k(\mathbf{X})Q_k(\mathbf{W})\|_F \\
&\leq \|\mathbf{X}(\mathbf{W} - Q_k(\mathbf{W}))\|_F + \|Q_k(\mathbf{W})(\mathbf{X} - Q_k(\mathbf{X}))\|_F \\
&\leq \|\mathbf{X}\|_F \|\mathbf{W} - Q_k(\mathbf{W})\|_F + \|Q_k(\mathbf{W})\|_F \|\mathbf{X} - Q_k(\mathbf{X})\|_F.
\end{aligned}$$

$\square$

### B.2 PROOF OF THEOREM 2

**Theorem 4.2.** The $k$-bit quantized weight error upper bound can be computed as follows:

$$||\mathbf{W} - Q_k(\mathbf{W})||_F \leq \frac{\max(|\mathbf{W}|)\sqrt{mn}}{2q_{\max}} \tag{10}$$

*Proof.*

$$\begin{aligned}
\|\mathbf{W} - Q_k(\mathbf{W})\|_F &= \left\| s_k^{\mathbf{W}} \cdot \frac{\mathbf{W}}{s_k^{\mathbf{W}}} - s_k^{\mathbf{W}} \cdot \text{round}\left(\frac{\mathbf{W}}{s_k^{\mathbf{W}}}\right) \right\|_F \\
&= |s_k^{\mathbf{W}}| \cdot \left\| \frac{\mathbf{W}}{s_k^{\mathbf{W}}} - \text{round}\left(\frac{\mathbf{W}}{s_k^{\mathbf{W}}}\right) \right\|_F \\
&\leq |s_k^{\mathbf{W}}| \cdot \sqrt{mn} \left\| \frac{\mathbf{W}}{s_k^{\mathbf{W}}} - \text{round}\left(\frac{\mathbf{W}}{s_k^{\mathbf{W}}}\right) \right\|_\infty \\
&\leq |s_k^{\mathbf{W}}| \cdot \frac{\sqrt{mn}}{2} = \frac{\max(|\mathbf{W}|)\sqrt{mn}}{2q_{\max}}.
\end{aligned}$$

The first inequality follows from the relationship between the Frobenius and infinity norms. For any matrix $\mathbf{W} \in \mathbb{R}^{m \times n}$, $\|\mathbf{W}\|_F = \left(\sum_{i,j}|w_{ij}|^2\right)^{1/2} \leq \left(\sum_{i,j}\|\mathbf{W}\|_\infty^2\right)^{1/2} = \sqrt{mn}\,\|\mathbf{W}\|_\infty$. The second inequality follows from the uniform rounding error bound. For each element $w_{ij}$ of $\mathbf{W}$, $\frac{w_{ij}}{s_k^{\mathbf{W}}} - \text{round}\left(\frac{w_{ij}}{s_k^{\mathbf{W}}}\right) \in \left[-\frac{1}{2}, \frac{1}{2}\right]$.

$\square$

### B.3 PROOF OF BIT-PLANE SLICING

For an arbitrary $K$-bit signed integer weight matrix $\mathbf{Q}_K^{\mathbf{W}}$ and any split index $0 < k_1 < K$, the bit-plane slicing can be written as:

$$\mathbf{Q}_K^{\mathbf{W}} = 2^{k_1} \cdot \mathbf{Q}_{k_2}^{\mathbf{W}} + \mathbf{U}_{k_1}^{\mathbf{W}}$$

where $\mathbf{U}_{k_1}^{\mathbf{W}}$ and $\mathbf{Q}_{k_2}^{\mathbf{W}}$ are the $k_1$-bit unsigned and $k_2$-bit signed representations of $\mathbf{W}$, respectively.

*Proof.* Let $\{\mathbf{B}^{(i)}\}_{i=0}^{K-1} \in \{0,1\}^{m \times n}$ denotes the $i$–th bit–plane of $\mathbf{Q}_K^{\mathbf{W}}$, thus

$$\begin{aligned}
\mathbf{Q}_K^{\mathbf{W}} &= -2^{K-1}\mathbf{B}^{(K-1)} + \sum_{i=0}^{K-2} 2^i \mathbf{B}^{(i)} \\
&= -2^{K-1}\mathbf{B}^{(K-1)} + \sum_{i=k_1}^{K-2} 2^i \mathbf{B}^{(i)} + \sum_{i=0}^{k_1-1} 2^i \mathbf{B}^{(i)} \\
&= 2^{k_1} \cdot \left( -2^{K-1-k_1}\mathbf{B}^{(K-1)} + \sum_{i=0}^{K-2-k_1} 2^i \mathbf{B}^{(i)} \right) + \sum_{i=0}^{k_1-1} 2^i \mathbf{B}^{(i)} \\
&= 2^{k_1} \cdot \left( -2^{k_2-1}\mathbf{B}^{(K-1)} + \sum_{i=0}^{k_2-2} 2^i \mathbf{B}^{(i)} \right) + \sum_{i=0}^{k_1-1} 2^i \mathbf{B}^{(i)} \\
&= 2^{k_1} \cdot \mathbf{Q}_{k_2}^{\mathbf{W}} + \mathbf{U}_{k_1}^{\mathbf{W}}.
\end{aligned}$$

$\square$

## B.4 PROOF OF THEOREM 3

**Theorem 4.3.** The upper bound of ZipperQuant error can be described as follows:

$$E(\hat{\mathbf{X}}, \hat{\mathbf{W}}) \leq \|\hat{\mathbf{X}}\|_F \|\hat{\mathbf{W}} - U_{k_1}(\hat{\mathbf{W}}) - 2^{k_1} Q_{k_2}(\hat{\mathbf{W}})\|_F + \|U_{k_1}(\hat{\mathbf{W}})\|_F \|\hat{\mathbf{X}} - Q_{k_1}(\hat{\mathbf{X}})\|_F. \quad (11)$$

*Proof.*

$$
\begin{aligned}
E(\hat{\mathbf{X}}, \hat{\mathbf{W}}) &= \left\| \hat{\mathbf{X}}\hat{\mathbf{W}} - Q_{k_1}(\hat{\mathbf{X}})U_{k_1}(\hat{\mathbf{W}}) - \hat{\mathbf{X}}\, 2^{k_1} Q_{k_2}(\hat{\mathbf{W}}) \right\|_F \\
&= \left\| \hat{\mathbf{X}}\hat{\mathbf{W}} - Q_{k_1}(\hat{\mathbf{X}})U_{k_1}(\hat{\mathbf{W}}) - \hat{\mathbf{X}}U_{k_1}(\hat{\mathbf{W}}) + \hat{\mathbf{X}}U_{k_1}(\hat{\mathbf{W}}) - \hat{\mathbf{X}}\, 2^{k_1} Q_{k_2}(\hat{\mathbf{W}}) \right\|_F \\
&= \left\| \hat{\mathbf{X}}\left( \hat{\mathbf{W}} - U_{k_1}(\hat{\mathbf{W}}) - 2^{k_1} Q_{k_2}(\hat{\mathbf{W}}) \right) + U_{k_1}(\hat{\mathbf{W}})\left( \hat{\mathbf{X}} - Q_{k_1}(\hat{\mathbf{X}}) \right) \right\|_F \\
&\leq \left\| \hat{\mathbf{X}}\left( \hat{\mathbf{W}} - U_{k_1}(\hat{\mathbf{W}}) - 2^{k_1} Q_{k_2}(\hat{\mathbf{W}}) \right) \right\| + \left\| U_{k_1}(\hat{\mathbf{W}})\left( \hat{\mathbf{X}} - Q_{k_1}(\hat{\mathbf{X}}) \right) \right\|_F \\
&\leq \left\| \hat{\mathbf{X}} \right\|_F \left\| \hat{\mathbf{W}} - U_{k_1}(\hat{\mathbf{W}}) - 2^{k_1} Q_{k_2}(\hat{\mathbf{W}}) \right\|_F + \left\| U_{k_1}(\hat{\mathbf{W}}) \right\|_F \left\| \hat{\mathbf{X}} - Q_{k_1}(\hat{\mathbf{X}}) \right\|_F
\end{aligned}
$$

$\square$

## C QUANTIZATION

### C.1 UNSIGNED INTEGER QUANTIZATION

For a non-negative tensor $\mathbf{X} \in \mathbb{R}^{t \times m}$, the $k$-bit unsigned symmetric quantization is defined as

$$\mathbf{U}_k^{\mathbf{X}} = \text{round}\left( \frac{\mathbf{X}}{s_k^{\mathbf{X}}} \right), \quad s_k^{\mathbf{X}} = \frac{\max(\mathbf{X})}{q_{\max}}, \quad (12)$$

where $\mathbf{U}_k^{\mathbf{X}}$ denotes the $k$-bit unsigned integer representation of $\mathbf{X}$, $s_k^{\mathbf{X}}$ is the scaling factor, and $q_{\max}$ is the maximum representable integer. For $k$-bit unsigned integers, $q_{\max} = 2^k - 1$. The corresponding dequantized tensor is $U_k(\mathbf{X}) = s_k^{\mathbf{X}} \mathbf{U}_k^{\mathbf{X}}$. For a linear layer with non-negative activation $\mathbf{X}$ and weight $\mathbf{W}$, the quantized computation can be expressed as

$$\mathbf{X}\mathbf{W} \approx U_k(\mathbf{X})\, U_k(\mathbf{W}) = s_k^{\mathbf{X}} s_k^{\mathbf{W}} \cdot \mathbf{U}_k^{\mathbf{X}} \mathbf{U}_k^{\mathbf{W}}. \quad (13)$$

## D BENCHMARK DETAILS

### D.1 BASELINES

We benchmark our methods compared with the following six baselines:

- RTN (Yao et al., 2023) employs uniform rounding-to-nearest with fixed scaling to quantize weights and activations without calibration, thereby offering a almost zero-cost baseline at the expense of larger accuracy loss.

- GPTQ (Frantar et al., 2023) employs post-training, Hessian-aware greedy weight quantization while applying error compensation to previously quantized rows, thereby preserving accuracy under W4A16.

- AWQ (Lin et al., 2024) employs activation-aware weight clipping and per-channel scaling while prioritizing salient channels, thereby achieving accurate W4A16 quantization with light calibration.

- SmoothQuant (Xiao, Guangxuan and Lin, Ji and Seznec, Mickael and Wu, Hao and Demouth, Julien and Han, Song, 2023b) employs channel-wise scaling to shift activation outliers into weights while smoothing activation distributions, thereby enabling W8A8 quantization with minimal degradation.

- QuaRot (Ashkboos et al., 2025) employs online Hadamard-based orthogonal rotations to on-the-fly disperse outliers and quantize in the rotated space, thereby enabling W4A4 quantization with minimal overhead and small loss.

- SpinQuant (Liu et al., 2025) employs learned orthogonal rotations to homogenize weights and activations magnitudes while reducing outliers, thereby enabling W4A4 quantization with improved stability.

# E    ADDITIONAL RESULTS

## E.1    IMPLEMENTATION DETAILS

We perform quantization on a single H20 with 96 GB of memory. ~~and inference is evaluated on a machine with one RTX 4090 (24 GB) and an Intel Xeon Gold 6430 with 120 GB of system memory~~ At inference time, we evaluate ZipperQuant on two GPU–CPU platforms including a common workstation RTX 4090 and a inference server L20 (see Section 5.1 for more details) . We implement INT4 multiplication on Tensor Cores using NVIDIA CUTLASS, and realize the CPU LUT with the AVX2 instruction set. For W4A8 quantization, we use per-token dynamic activation quantization and per-channel weight quantization. For the 4-bit setting, we adopt per-group and per-channel symmetric quantization for activations and weights, respectively. INT4 quantization uses a group size of 64 with 16-bit scales. For ZipperQuant setting, we choose $K = 6$, $k_1 = 4$ and $k_2 = 2$.

The smoothing factor $\boldsymbol{\lambda} \in \mathbb{R}^m$ is a per-channel vector whose $i$-th element is computed as $\lambda_i = \max(|\mathbf{X}_{:,i}|)^\alpha / \max(|\mathbf{W}_{i,:}|)^{1-\alpha}$ where $\mathbf{X} \in \mathbb{R}^{t \times m}$ and $\mathbf{W} \in \mathbb{R}^{m \times n}$ following SmoothQuant (Xiao, Guangxuan and Lin, Ji and Seznec, Mickael and Wu, Hao and Demouth, Julien and Han, Song, 2023b). The migration strength $\alpha$ is chosen offline, per layer, by searching for the value that minimizes the layer output mean squared error (MSE) after ZipperQuant on the calibration dataset.

For the baselines, we run AWQ (Lin et al., 2024), FlatQuant (Sun et al., 2025), QuaRot (Ashkboos et al., 2025), and QServe (Lin et al., 2025) using their publicly released implementations. For AWQ, FlatQuant, and QuaRot, we adopt the official PyTorch-based code and recommended W4A4/W4A8 configurations for each model, and ensure that all methods share the same decoding setup (greedy decoding), batch sizes, and sequence lengths on the same GPU. For QServe, we build its C++/CUDA runtime following the official instructions and evaluate the W4A8 configuration on the same hardware and workload as ZipperQuant. All throughput measurements are collected after warm-up runs to avoid initialization overhead.

## E.2    ZERO-SHOT TASK RESULTS

In Table 5 and Table 6, we show the complete results of Table 1. including ARC-Challenge, ARC-Easy (Clark et al., 2018), HellaSwag (Zellers et al., 2019), LAMBADA (Paperno et al., 2016), PIQA(Bisk et al., 2020), and WinoGrande (Sakaguchi et al., 2021). Additionally, we also report the perplexity score on WikiText2 (Merity et al., 2017) datasets. We compare our results with previous works including RTN (Yao et al., 2023), GPTQ (Frantar et al., 2023), AWQ (Lin et al., 2024), SmoothQuant (Xiao, Guangxuan and Lin, Ji and Seznec, Mickael and Wu, Hao and Demouth, Julien and Han, Song, 2023b), QuaRot (Ashkboos et al., 2025) and SpinQuant (Liu et al., 2025).

Table 5: Complete comparison of the perplexity score on WikiText2 and averaged accuracy on six zero-shot tasks on LLaMA3 1B/3B/8B.

| Model | Precision | Method | ARC-C | ARC-E | HellaSwag | PIQA | Winogrande | LAMBADA | Avg. | Wiki2 ($\downarrow$) |
|---|---|---|---|---|---|---|---|---|---|---|
| | W16A16 | – | 38.7 | 65.2 | 60.7 | 75.3 | 60.9 | 59.6 | 60.1 | 13.4 |
| | W4A16 | RTN | 40.0 | 62.6 | 56.6 | 72.2 | 59.9 | 57.8 | 58.2 | 20.7 |
| | W4A16 | GPTQ | 38.5 | 61.7 | 56.4 | 71.4 | 58.7 | 59.3 | 57.6 | 17.3 |
| | W4A16 | AWQ | 38.4 | 63.6 | 56.4 | 72.5 | 57.9 | 58.7 | 57.9 | 15.6 |
| 1B | W4A8 | RTN | 38.7 | 62.4 | 56.6 | 72.1 | 58.6 | 50.6 | 56.5 | 20.7 |
| | W4A8 | SmoothQuant | 30.7 | 47.6 | 47.6 | 64.9 | 52.9 | 50.8 | 49.1 | 108.2 |
| | W4A8 | QuaRot | 37.4 | 58.7 | 57.3 | 68.9 | 55.5 | 56.2 | 55.7 | 18.9 |
| | W4A8 | SpinQuant | 39.5 | 60.8 | 57.7 | 73.2 | 60.3 | 57.4 | 58.2 | 15.3 |
| | W4A8 | ZipperQuant | 38.6 | 63.6 | 57.9 | 73.7 | 60.5 | 58.9 | 58.9 | 15.1 |
| | W4A4 | RTN | 28.3 | 37.8 | 35.9 | 56.4 | 51.4 | 45.8 | 42.6 | 137.5 |
| | W4A4 | SmoothQuant | 26.4 | 32.3 | 28.7 | 54.7 | 48.0 | 43.6 | 38.9 | 2027.8 |
| | W4A4 | QuaRot | 28.4 | 59.1 | 45.6 | 60.3 | 49.8 | 50.5 | 48.9 | 50.3 |
| | W4A4 | SpinQuant | 29.7 | 59.2 | 51.0 | 69.7 | 50.8 | 51.7 | 52.0 | 44.8 |
| | W4A4 | ZipperQuant | 32.8 | 60.4 | 53.2 | 69.8 | 53.8 | 55.0 | 54.2 | 28.9 |
| | W16A16 | – | 47.6 | 69.9 | 71.0 | 76.0 | 66.6 | 65.9 | 66.2 | 10.7 |
| | W4A16 | RTN | 41.3 | 60.2 | 66.2 | 73.1 | 63.0 | 61.6 | 60.9 | 18.8 |
| | W4A16 | GPTQ | 41.4 | 60.8 | 65.9 | 73.6 | 65.0 | 63.4 | 61.7 | 15.2 |
| | W4A16 | AWQ | 43.5 | 66.7 | 65.8 | 75.8 | 62.7 | 63.8 | 63.0 | 12.7 |
| 3B | W4A8 | RTN | 42.6 | 60.2 | 66.2 | 72.6 | 62.7 | 60.1 | 60.7 | 29.0 |
| | W4A8 | SmoothQuant | 40.7 | 59.5 | 65.5 | 73.8 | 58.5 | 60.7 | 59.8 | 288.5 |
| | W4A8 | QuaRot | 42.9 | 64.3 | 68.1 | 72.6 | 64.8 | 61.2 | 62.3 | 12.4 |
| | W4A8 | SpinQuant | 44.2 | 65.9 | 68.3 | 74.8 | 65.9 | 63.8 | 63.8 | 11.6 |
| | W4A8 | ZipperQuant | 46.2 | 65.8 | 69.2 | 75.1 | 65.7 | 63.9 | 64.3 | 11.1 |
| | W4A4 | RTN | 29.8 | 41.0 | 41.4 | 57.3 | 50.9 | 38.9 | 43.2 | 741.9 |
| | W4A4 | SmoothQuant | 30.5 | 43.6 | 37.7 | 58.0 | 52.9 | 45.3 | 44.7 | 372.3 |
| | W4A4 | QuaRot | 40.6 | 49.9 | 51.2 | 61.9 | 61.4 | 55.2 | 53.4 | 26.9 |
| | W4A4 | SpinQuant | 43.7 | 54.6 | 56.3 | 66.7 | 65.5 | 57.9 | 57.5 | 22.4 |
| | W4A4 | ZipperQuant | 45.8 | 58.1 | 63.5 | 70.3 | 64.5 | 61.6 | 60.6 | 20.6 |
| | W16A16 | – | 57.7 | 77.6 | 79.6 | 80.7 | 73.7 | 75.6 | 74.2 | 6.1 |
| | W4A16 | RTN | 48.1 | 73.7 | 75.5 | 77.1 | 73.4 | 70.2 | 69.7 | 8.2 |
| | W4A16 | GPTQ | 46.8 | 71.5 | 73.9 | 76.6 | 70.7 | 72.3 | 68.6 | 7.2 |
| | W4A16 | AWQ | 48.3 | 74.6 | 78.0 | 79.3 | 73.3 | 73.5 | 71.2 | 7.3 |
| 8B | W4A8 | RTN | 48.1 | 73.2 | 75.5 | 77.1 | 72.5 | 60.7 | 67.9 | 8.2 |
| | W4A8 | SmoothQuant | 41.0 | 67.5 | 70.8 | 74.9 | 69.1 | 63.2 | 64.4 | 10.7 |
| | W4A8 | QuaRot | 43.3 | 68.0 | 72.9 | 75.1 | 65.8 | 65.8 | 65.2 | 7.8 |
| | W4A8 | SpinQuant | 54.0 | 76.5 | 77.5 | 79.6 | 72.4 | 72.4 | 72.1 | 6.7 |
| | W4A8 | ZipperQuant | 54.8 | 75.5 | 77.9 | 79.2 | 73.8 | 73.9 | 72.5 | 6.6 |
| | W4A4 | RTN | 29.5 | 42.7 | 41.2 | 57.8 | 49.4 | 51.6 | 45.4 | 241.6 |
| | W4A4 | SmoothQuant | 26.3 | 36.3 | 31.4 | 52.9 | 34.6 | 45.7 | 37.9 | 867.5 |
| | W4A4 | QuaRot | 41.6 | 62.0 | 64.9 | 75.1 | 62.3 | 63.3 | 61.5 | 21.4 |
| | W4A4 | SpinQuant | 45.9 | 66.5 | 75.9 | 68.0 | 68.5 | 67.9 | 65.5 | 18.6 |
| | W4A4 | ZipperQuant | 47.9 | 69.2 | 73.5 | 73.6 | 71.3 | 70.1 | 67.6 | 16.5 |

Table 6: Complete comparison of the perplexity score on WikiText2 and averaged accuracy on six zero-shot tasks on Qwen3 4B/8B/14B/32B.

| Model | Precision | Method | ARC-C | ARC-E | HellaSwag | PIQA | Winogrande | LAMBADA | Avg. | Wiki2 ($\downarrow$) |
|---|---|---|---|---|---|---|---|---|---|---|
| 4B | W16A16 | – | 50.6 | 80.5 | 52.2 | 75.0 | 65.8 | 59.4 | 63.9 | 13.7 |
| | W4A16 | RTN | 44.7 | 75.3 | 48.6 | 73.2 | 61.8 | 57.9 | 60.2 | 17.6 |
| | W4A16 | GPTQ | 45.4 | 76.6 | 50.1 | 74.4 | 63.9 | 58.2 | 61.4 | 14.5 |
| | W4A16 | AWQ | 46.2 | 76.6 | 50.1 | 73.8 | 63.1 | 58.5 | 61.4 | 16.6 |
| | W4A8 | RTN | 42.3 | 68.7 | 45.8 | 68.8 | 53.9 | 55.1 | 55.8 | 30.6 |
| | W4A8 | SmoothQuant | 42.2 | 69.5 | 46.1 | 71.5 | 57.5 | 55.0 | 57.0 | 22.6 |
| | W4A8 | QuaRot | 43.3 | 70.5 | 48.3 | 71.7 | 57.9 | 55.4 | 57.8 | 16.7 |
| | W4A8 | SpinQuant | 44.2 | 70.2 | 48.1 | 70.9 | 57.4 | 54.8 | 57.6 | 17.1 |
| | W4A8 | ZipperQuant | 46.3 | 74.8 | 50.3 | 72.8 | 61.5 | 55.0 | 60.1 | 15.3 |
| | W4A4 | RTN | 26.7 | 28.6 | 24.7 | 48.9 | 51.8 | 48.9 | 38.3 | 8791 |
| | W4A4 | SmoothQuant | 22.6 | 25.9 | 25.6 | 51.6 | 47.9 | 50.2 | 37.3 | 9910 |
| | W4A4 | QuaRot | 39.8 | 65.7 | 44.3 | 62.5 | 54.6 | 52.4 | 53.2 | 21.5 |
| | W4A4 | SpinQuant | 40.9 | 66.9 | 45.7 | 61.9 | 54.9 | 55.0 | 54.2 | 21.7 |
| | W4A4 | ZipperQuant | 43.6 | 70.4 | 48.1 | 65.0 | 60.2 | 54.6 | 57.0 | 18.9 |
| 8B | W16A16 | – | 55.5 | 83.5 | 57.1 | 76.4 | 68.0 | 67.4 | – | 9.71 |
| | W4A16 | RTN | 53.8 | 79.1 | 53.8 | 75.4 | 63.6 | 60.6 | 64.4 | 12.0 |
| | W4A16 | GPTQ | 55.6 | 80.1 | 55.6 | 76.0 | 66.1 | 62.8 | 66.0 | 10.3 |
| | W4A16 | AWQ | 59.7 | 83.1 | 59.7 | 79.4 | 72.0 | 62.7 | 69.4 | 10.5 |
| | W4A8 | RTN | 45.8 | 73.4 | 50.8 | 73.1 | 62.9 | 58.7 | 60.8 | 12.3 |
| | W4A8 | SmoothQuant | 44.3 | 71.0 | 51.7 | 73.4 | 62.1 | 60.0 | 60.8 | 12.5 |
| | W4A8 | QuaRot | 53.9 | 77.8 | 51.9 | 72.7 | 62.6 | 60.5 | 63.2 | 12.9 |
| | W4A8 | SpinQuant | 54.2 | 79.2 | 52.0 | 72.7 | 63.8 | 61.3 | 63.9 | 12.1 |
| | W4A8 | ZipperQuant | 53.8 | 79.0 | 54.1 | 74.6 | 63.5 | 62.8 | 64.6 | 11.5 |
| | W4A4 | RTN | 22.6 | 24.9 | 28.0 | 48.9 | 51.8 | 46.9 | 37.2 | 4392 |
| | W4A4 | SmoothQuant | 25.7 | 25.5 | 26.7 | 50.5 | 52.2 | 44.9 | 37.6 | 3360.1 |
| | W4A4 | QuaRot | 49.8 | 74.8 | 49.8 | 68.3 | 60.7 | 57.1 | 60.1 | 24.5 |
| | W4A4 | SpinQuant | 50.2 | 75.6 | 50.6 | 68.3 | 61.1 | 58.0 | 60.6 | 25.9 |
| | W4A4 | ZipperQuant | 51.8 | 77.2 | 52.0 | 70.9 | 61.4 | 58.7 | 62.0 | 21.4 |
| 14B | W16A16 | – | 59.0 | 84.3 | 60.9 | 80.0 | 72.9 | 68.4 | 70.9 | 8.6 |
| | W4A16 | RTN | 51.8 | 80.6 | 58.9 | 77.9 | 68.7 | 64.4 | 67.0 | 9.9 |
| | W4A16 | GPTQ | 57.1 | 81.9 | 59.6 | 78.8 | 72.5 | 66.9 | 69.5 | 9.2 |
| | W4A16 | AWQ | 51.7 | 79.2 | 55.6 | 76.0 | 66.1 | 66.5 | 65.9 | 9.6 |
| | W4A8 | RTN | 50.7 | 79.5 | 57.2 | 75.3 | 66.9 | 62.0 | 65.3 | 10.9 |
| | W4A8 | SmoothQuant | 50.5 | 79.0 | 57.0 | 75.8 | 66.3 | 61.3 | 65 | 11.0 |
| | W4A8 | QuaRot | 56.4 | 80.7 | 57.6 | 76.9 | 68.5 | 65.8 | 67.6 | 10.4 |
| | W4A8 | SpinQuant | 56.2 | 81.2 | 57.6 | 77.2 | 68.9 | 66.3 | 67.9 | 10.2 |
| | W4A8 | ZipperQuant | 55.4 | 82.4 | 58.5 | 78.3 | 69.0 | 66.4 | 68.3 | 9.9 |
| | W4A4 | RTN | 24.8 | 23.9 | 26.9 | 55.7 | 48.6 | 50.7 | 38.4 | 18749 |
| | W4A4 | SmoothQuant | 26.5 | 25.8 | 25.8 | 51.2 | 50.2 | 45.3 | 37.5 | 21675 |
| | W4A4 | QuaRot | 52.8 | 76.4 | 54.1 | 73.6 | 65.0 | 62.8 | 64.1 | 18.2 |
| | W4A4 | SpinQuant | 53.2 | 77.8 | 54.6 | 72.9 | 65.5 | 63.4 | 64.6 | 17.5 |
| | W4A4 | ZipperQuant | 53.1 | 78.9 | 56.6 | 76.2 | 66.0 | 65.5 | 66.0 | 12.0 |
| 32B | W16A16 | – | 57.8 | 84.4 | 63.9 | 80.9 | 73.6 | 70.3 | 71.8 | 7.6 |
| | W4A16 | RTN | 49.7 | 73.0 | 44.4 | 71.9 | 62.9 | 68.5 | 61.7 | 38.5 |
| | W4A16 | GPTQ | 57.8 | 82.6 | 62.7 | 79.7 | 67.6 | 69.8 | 70.0 | 8.3 |
| | W4A16 | AWQ | 56.9 | 82.5 | 63.1 | 79.8 | 71.8 | 68.7 | 70.5 | 8.2 |
| | W4A8 | RTN | 49.5 | 76.3 | 60.7 | 72.9 | 65.3 | 62.0 | 64.4 | 11.2 |
| | W4A8 | SmoothQuant | 49.2 | 75.8 | 61.2 | 71.4 | 63.5 | 63.7 | 64.1 | 11.6 |
| | W4A8 | QuaRot | 51.3 | 78.9 | 61.0 | 76.8 | 70.9 | 68.5 | 67.9 | 10.5 |
| | W4A8 | SpinQuant | 51.7 | 79.2 | 61.2 | 76.9 | 71.0 | 69.1 | 68.2 | 10.3 |
| | W4A8 | ZipperQuant | 55.3 | 79.8 | 62.6 | 76.5 | 71.6 | 68.7 | 69.1 | 9.5 |
| | W4A4 | RTN | 28.5 | 27.6 | 30.4 | 52.5 | 50.7 | 52.0 | 40.3 | 1796 |
| | W4A4 | SmoothQuant | 26.0 | 28.9 | 31.5 | 51.9 | 54.1 | 48.6 | 40.2 | 1806 |
| | W4A4 | QuaRot | 49.8 | 72.2 | 58.9 | 73.4 | 67.9 | 65.3 | 64.6 | 15.6 |
| | W4A4 | SpinQuant | 50.3 | 73.6 | 59.3 | 72.8 | 67.3 | 66.0 | 64.9 | 16.0 |
| | W4A4 | ZipperQuant | 52.6 | 75.3 | 60.4 | 73.4 | 69.4 | 65.7 | 66.1 | 12.4 |

## E.3 ADDITIONAL THROUGHPUT ON AN L20 INFERENCE SERVER

Table 7 reports end-to-end decoding throughput on an inference server equipped with an NVIDIA L20 GPU and an Intel Xeon Platinum 8457C CPU (see Section 5.1 for details). We evaluate TensorRT-LLM (FP16) and several W4A4 methods under batch sizes 1–8 with greedy decoding.

Across all batch sizes, ZipperQuant remains the fastest W4A4 method on the L20 platform, yielding roughly 3–7× higher throughput than FlatQuant and QuaRot. Compared to the full-precision TensorRT-LLM baseline, ZipperQuant delivers about 1.3–1.7× speedup, and we observe the same favorable trend against the AWQ baseline as on the RTX 4090 (AWQ numbers on L20 are omitted from the table for brevity). These results indicate that our speed–accuracy trade-off is stable across different GPU and CPU platforms. Our CPU-side Zipper engine is implemented using Intel AVX2 intrinsics, and thus applies to any Intel CPU that supports AVX2.

Table 7: End-to-end decoding throughput (tokens/s) on an inference server with an NVIDIA L20 GPU. All methods are evaluated on the same model with greedy decoding.

| Method | Batch size=1 | Batch size=2 | Batch size=4 | Batch size=8 |
|---|---|---|---|---|
| TensorRT-LLM (W16A16) | 47.80 | 95.30 | 190.40 | 375.90 |
| FlatQuant (W4A4) | 18.80 | 33.57 | 63.50 | 143.39 |
| QuaRot (W4A4) | 11.83 | 21.68 | 40.52 | 192.45 |
| ZipperQuant (W4A4) | 81.62 | 141.86 | 237.37 | 622.93 |

## E.4 LATENCY BREAKDOWN OF THE CPU–GPU PIPELINE

To understand the cost of the CPU path and PCIe transfers, we measure a detailed latency breakdown on an NVIDIA RTX 4090 for LLaMA3-8B. Table 8 reports microsecond-level latency for representative linear layers (q_proj, k_proj, v_proj, o_proj, down_proj) under different batch sizes. We decompose each step into GPU INT4 GEMM, GPU-side activation quantization, CPU-side device-to-host (D2H) transfer, CPU LUT computation, and host-to-device (H2D) transfer.

In our implementation, activation quantization and D2H transfers are pipelined with GPU computation: once the activations are quantized on the GPU, they are streamed to the CPU while the GPU proceeds with subsequent work. As the table shows, CPU LUT computation and PCIe transfers account for a comparable amount of time as the GPU INT4 GEMM, but these costs are largely overlapped in the end-to-end pipeline. Consequently, the CPU path does not dominate the total latency and enables ZipperQuant to maintain strong throughput, as reported in Section 5.3.

Table 8: Latency breakdown (in $\mu$s) on an NVIDIA RTX 4090 for LLaMA3-8B. We report representative self-attention and FFN linear layers under different batch sizes.

| Batch size | Layer | GPU INT4 GEMM | GPU Quantize | GPU Total | CPU D2H | CPU LUT | CPU H2D | CPU Total |
|---|---|---|---|---|---|---|---|---|
| 1 | q_proj, o_proj | 116 | 78 | 194.0 | 16.0 | 165.0 | 17.0 | 198.1 |
|   | k_proj, v_proj | 112 | 31 | 143.0 | 15.3 | 113.0 | 15.5 | 143.7 |
|   | down_proj | 170 | 123 | 293.0 | 22.6 | 249.0 | 23.3 | 295.0 |
| 2 | q_proj, o_proj | 124 | 103 | 227.0 | 17.1 | 201.3 | 17.5 | 235.9 |
|   | k_proj, v_proj | 114 | 51 | 165.0 | 15.5 | 136.7 | 15.8 | 168.1 |
|   | down_proj | 269 | 213 | 482.0 | 22.3 | 423.3 | 22.5 | 468.1 |
| 4 | q_proj, o_proj | 173 | 201 | 374.0 | 19.2 | 330.0 | 19.7 | 368.9 |
|   | k_proj, v_proj | 115 | 65 | 180.0 | 16.0 | 149.2 | 16.5 | 181.7 |
|   | down_proj | 452 | 312 | 764.0 | 30.6 | 522.9 | 30.2 | 583.7 |
| 8 | q_proj, o_proj | 318 | 285 | 603.0 | 24.3 | 528.0 | 23.1 | 575.5 |
|   | k_proj, v_proj | 127 | 93 | 220.0 | 18.1 | 196.9 | 18.8 | 233.8 |
|   | down_proj | 801 | 531 | 1332.0 | 46.2 | 1045.8 | 46.7 | 1138.6 |

### E.5 SPARSITY FIGURE

We further analyze the sparsity patterns of the projection matrices in both the self-attention and MLP blocks. Specifically, we report the per-layer sparsity of the highest and the second-highest INT groups for the self-attention q/k/v projections and the MLP up/gate/down projections. As shown in Fig. 8, all six matrices exhibit extremely high sparsity in the highest INT group, while the second-highest group also remains highly sparse across layers.

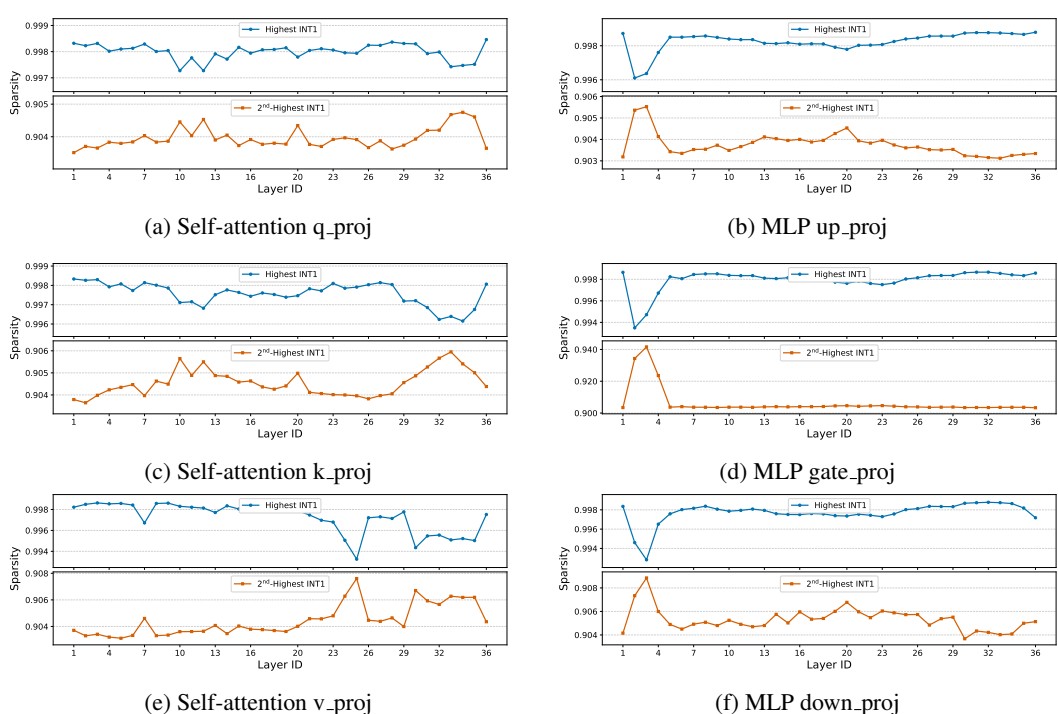

Figure 8: Weight sparsity patterns of self-attention (q/k/v) and MLP (up/gate/down) projection matrices.

