# OpenReview forum: "ZipperQuant: Bit-Based Inlier–Outlier Disaggregation for 4-Bit LLMs on GPU–CPU"
_ICLR.cc/2026/Conference — Submitted to ICLR 2026_

### Official Review · Reviewer_QWVN · 2025-11-01

**Soundness:** 3
**Presentation:** 3
**Contribution:** 3
**Rating:** 4
**Confidence:** 3

**Summary:**

The paper introduces **ZipperQuant**, a 4-bit quantization paradigm for Large Language Models (LLMs). It addresses the significant accuracy degradation that occurs in low-bit quantization due to the challenge of simultaneously representing both high-frequency "inlier" values and rare, large-magnitude "outlier" values. The proposed solution is a hybrid GPU-CPU approach based on **bit-based disaggregation**. The method first uses smoothing to migrate activation outliers into the weights. Then, these weights are decomposed into low-order and high-order bit components. The GPU executes the dense computations, (all inliers and the low-order bits of outliers) using low-precision INT4 units. Concurrently, the sparse high-order bits are offloaded to the CPU for high-precision computation. To mitigate the performance gap between the GPU and CPU, the authors develop a specialized **lookup-table (LUT) mechanism** that accelerates the CPU-side computation by replacing multiplications with efficient table lookups and accumulation. The primary contributions are this bit-based disaggregation strategy, the co-designed Zipper engine with its LUT-based acceleration, and experimental results showing that the method achieves near-FP16 accuracy for W4A4 quantization, along with significant speedups (up to 3.01x) over W4A16 baselines.

**Strengths:**

* **Novel and well-motivated disaggregation strategy**
    * The paper clearly articulates the problem with naive *value-based* offloading, citing the 10-100x GPU-CPU performance gap and PCIe transfer bottlenecks as critical flaws. This provides a strong motivation for a new approach.
    * The core idea of *bit-based* disaggregation is novel. It intelligently separates the computation, keeping the dense, low-order bits on the GPU to leverage low-precision hardware (INT4 GEMM) while moving only the sparse, high-order bits to the CPU for high-precision handling. This is a clever and practical way to balance precision requirements with computational efficiency.
    * This design choice is empirically justified by analysis showing that the high-order bits of weights are indeed highly sparse across layers (Figure 3b), making them suitable for the proposed sparse computation on the CPU.
    * The proposal is not purely theoretical; it includes a "Zipper engine" and a detailed LUT-based workflow (Figure 4b), demonstrating a clear and practical path to implementation.

* **Strong empirical results for W4A4 quantization**
    * The primary accuracy results in Table 1 are compelling.ZipperQuant (W4A4) consistently and significantly outperforms other W4A4 methods, including RTN, SmoothQuant, QuaRot, and SpinQuant, in terms of both WikiText-2 perplexity and average zero-shot accuracy.
    * In many cases, the W4A4 ZipperQuant results are competitive with or even exceed W4A8 and W4A16 baselines (Table 1). For instance, on LLaMA-3.2 3B, ZipperQuant (W4A4) achieves a 60.6 average accuracy, closing much of the gap to the W16A16 baseline (66.2) and outperforming W4A4 SpinQuant (57.5).
    * The method is shown to be complementary to other SOTA techniques. Table 2 demonstrates that integrating the Zipper engine with SpinQuant ("Spin+Zipper") further improves accuracy over both methods individually. This suggests the bit-based disaggregation concept has broad applicability.

* **Comprehensive performance analysis and ablation studies**
    * The paper provides a detailed end-to-end speedup analysis for both prefill and decoding (Figure 5). The reported speedups over a W4A16 baseline (up to 3.01x prefill, 1.90x decoding) are significant and demonstrate the practical benefit of enabling W4A4 computation without sacrificing accuracy.
    * The ablation study on the CPU computation strategy (Figure 7) is essential. It provides clear evidence that a "Naive GEMM" implementation on the CPU is prohibitively slow (up to 101x slowdown) and validates that the proposed sparse LUT mechanism is critical for achieving the reported performance.
    * The accuracy ablations in Table 3 effectively justify the complete pipeline. They show that "Naive 4-bits" and "Smoothing" alone are insufficient, and critically, that applying ZipperQuant *without* smoothing also fails. This confirms that the combination of smoothing and bit-disaggregation is necessary.
    * The inclusion of memory footprint analysis (Figure 6) confirms the practical memory-saving benefits of the W4A4 approach compared to W4A16 and FP16 models.

**Weaknesses:**

* **Clarity and correctness of mathematical formulations**
    * The proof for Theorem 4.3 (ZipperQuant error bound) in Appendix B.4 appears to be inconsistent with the main text. The proof steps use $Q_{k_1}(\hat{W})$ and $Q_{k_2}(\hat{W})$, but the theorem statement in Equation 8 and Equation 11 relies on $U_{k_1}(\hat{W})$ and $2^{k_1}Q_{k_2}(\hat{W})$. Furthermore, the first line of the proof defines the error relative to a decomposition that does not seem to match the one proposed in Equation 7. This discrepancy makes it difficult to verify the correctness of the theoretical error bound.
    * The notation in Equation 6 is ambiguous. It defines $\aleph_{K}(\hat{W}) = s_K^W Q_K^W = \dots = U_{k_1}(\hat{W}) + 2^{k_1}Q_{k_2}(\hat{W})$. This structure implies that both the low-order unsigned component $U_{k_1}(\hat{W})$ and the high-order signed component $Q_{k_2}(\hat{W})$ are derived using the *same* scaling factor $s_K^W$ (from the K-bit quantization). This seems counterintuitive, and it's not clear how $U_{k_1}(\hat{W})$ is defined. This ambiguity obscures the precise mathematical definition of the decomposition.
    * The notation for quantized activations in the proof of Theorem 4.3 is confusing. The proof introduces $Q_{k_1}(\hat{X})$ and even $Q_{k_2}(\hat{X})$, but the (k1, k2) bit-slicing scheme is defined for weights, not activations. This makes the proof difficult to follow and its validity unclear.

* **Ambiguity in implementation details**
    * The paper states the bit split is $K=6$, $k_1=4$ (GPU), and $k_2=2$ (CPU). However, Figure 3c illustrates the split as "Highest INT1" (CPU), "2nd Highest INT1" (CPU), and "UINT4" (GPU). It is not clear how these two sparse 1-bit components are combined and processed as the single $k_2=2$ bit signed term ($Q_{k_2}^W$) described in Equations 6-7.
    * The "sign restoration" mechanism is critical but confusingly explained. The text mentions "subtracting 16 at those positions on the GPU" for negative entries, which is non-trivial for a UINT4 representation. The relationship between this step and the "dividing GPU weights by 2" step mentioned immediately after is also unclear.
    * The LUT mechanism's performance (Figure 4b) depends on the group size $g$ (example $g=3$ is given). This hyperparameter, which dictates the LUT size ($2^g-1$) and computational trade-off, is not specified for the main experiments nor is its impact ablated.
    * Equation 7 implies a potential overhead: the GPU computes $Q_{k_1}(\hat{X})U_{k_1}(\hat{W})$ (using quantized activations) while the CPU computes $\hat{X} \cdot 2^{k_1}Q_{k_2}(\hat{W})$ (using high-precision activations $\hat{X}$). This suggests activations $\hat{X}$ must be quantized for the GPU path *and* simultaneously sent in high precision to the CPU for the LUT. This cost is not explicitly analyzed.

* **Limited performance comparisons and sensitivity analysis**
    * The main speedup comparisons (Figure 5) are against W16A16 and W4A16 (AWQ) baselines. While this is useful, the paper's core contribution is a W4A4 method. Direct, quantitative speedup comparisons against the other SOTA W4A4 methods (QuaRot, SpinQuant) are missing. The paper notes their "online conversion overhead" but does not provide data to assess the practical performance trade-off.
    * The latency analysis in Figure 4a is shown for "QKV projection". It is not guaranteed that this favorable latency holds for the much larger FFN layers. The paper does not provide a layer-wise latency breakdown to confirm that the CPU path does not become a bottleneck elsewhere.
    * The method's efficiency hinges on the assumption of high sparsity for high-order bits. Figure 3b shows this for K-projection in one model. This assumption needs to be validated more broadly (e.g., for FFN layers, other models). The paper lacks a sensitivity analysis showing how performance (latency, data transfer) degrades if this sparsity assumption weakens (e.g., if sparsity drops from >99.5% to 95% or 90%).
    * All experiments are on an RTX 4090. The performance of a hybrid GPU-CPU system is highly dependent on the specific CPU, GPU, and interconnect (PCIe). The conclusions drawn may not generalize to server-grade systems (e.g., H100 with NVLink) or on-device platforms with integrated memory, which have very different performance characteristics.

**Questions:**

* **Regarding mathematical formulations:**
    * Could the authors please clarify the derivation in Appendix B.4? Specifically, how does the proof correctly map to the decomposition in Equation 7 and the error bound in Equation 8, given the apparent mismatches in terms?
    * Could the authors please elaborate on the scaling factors in Equation 6? Are the low-order $U_{k_1}(\hat{W})$ and high-order $Q_{k_2}(\hat{W})$ components truly scaled by the same factor $s_K^W$? If not, could the notation be corrected to reflect the actual implementation?
    * Could the authors please review the notation for activations in the proof in Appendix B.4 (e.g., $Q_{k_2}(\hat{X})$) and clarify its meaning, as the k1/k2 split is defined for weights?

* **Regarding implementation details:**
    * Could the authors please explain how the $k_2=2$ bits for the CPU are implemented? Specifically, how are the "Highest INT1" and "2nd Highest INT1" (from Figure 3c) combined and processed to match the single $Q_{k_2}(\hat{W})$ term in Equation 7?
    * Could the authors provide a more concrete explanation or pseudo-code for the "sign restoration" mechanism? How is "subtracting 16" implemented on a UINT4 GPU value, and what is its precise interaction with the "dividing by 2" step?
    * What was the group size $g$ used for the LUT in the main experiments? Could the authors provide a sensitivity analysis on how $g$ affects end-to-end latency?
    * Does the ZipperQuant pipeline indeed require both quantizing $\hat{X}$ for the GPU and sending the full-precision $\hat{X}$ to the CPU, as implied by Equation 7? If so, what is the measured overhead of this dual requirement?

* **Regarding performance and bottlenecks:**
    * Could the authors provide a direct performance (latency/throughput) comparison against other W4A4 methods like QuaRot and SpinQuant, even if it requires discussing the impact of their respective overheads?
    * Does the low latency of the CPU path, shown for QKV projections in Figure 4a, also hold for the larger FFN layers? Can the authors confirm the CPU offload does not become a bottleneck in any part of the models tested?
    * How robust is the method's performance to the high-order bit sparsity assumption? Does this high sparsity (Figure 3b) hold for FFN layers as well? What is the performance impact if sparsity decreases?
    * Do the authors have any insights or preliminary data on how this GPU-CPU disaggregation strategy might perform on different hardware architectures, such as server-grade GPUs or edge devices?

---

> ### Author Response · Authors · 2025-11-26
> **Author Response to Reviewer QWVN (Part 1)**
>
> >**Question1-The proof for Theorem 4.3 (ZipperQuant error bound) in Appendix~B.4 appears to be inconsistent with the main text. The proof steps use $Q_{k_1}(\hat{\mathbf{W}})$ and $Q_{k_2}(\hat{\mathbf{W}})$, but the theorem statement in Equation 8 and Equation 11 relies on $U_{k_1}(\hat{\mathbf{W}})$ and $2^{k_1}Q_{k_2}(\hat{\mathbf{W}})$. Furthermore, the first line of the proof defines the error relative to a decomposition that does not seem to match the one proposed in Equation 7. This discrepancy makes it difficult to verify the correctness of the theoretical error bound.**
>
> Thank you for pointing out this notational discrepancy. The main text of Theorem 4.3 is correct, while there is a typo in the proof in Appendix B.4.
>
> *1-Clarification of notation in Appendix B.4.*
>
> The notation in Appendix B.4 is inconsistent with that in the main text. In Appendix B.4, we reused the symbols $Q_{k_1}(\hat{\mathbf{W}})$ and $Q_{k_2}(\hat{\mathbf{W}})$. They were intended to be shorthands for $U_{k_1}(\hat{\mathbf{W}})$ and $2^{k_1}Q_{k_2}(\hat{\mathbf{W}})$, respectively, as defined in the main text.
>
> *2-Consistency with Equation 8 and Theorem 4.3.*
>
>  Under this identification, the error term in the first line of Appendix B.4 is exactly
> \begin{equation}
> || \hat{\mathbf{X}} \hat{\mathbf{W}} - Q_{k_1}(\hat{\mathbf{X}}) U_{k_1}(\hat{\mathbf{W}}) - \hat{\mathbf{X}} 2^{k_1} Q_{k_2}(\hat{\mathbf{W}}) ||_F,
> \end{equation}
> which is the same GPU–CPU decomposition as in Equation 7. The subsequent steps then follow the same triangle-inequality argument as in Theorem 4.1 and lead to the bound stated in Theorem 4.3 (Equation 8). Therefore, the theoretical error bound and its conclusion remain valid; the issue is purely notational.
>
> In the revised version, we will keep the original notations in the main text and correct the notions in Appendix B.4 to avoid confusion.
>
> >**Question2-The notation in Equation 6 is ambiguous. It defines $Q_K(\hat{\mathbf{W}})$ = $s_{K}^{\mathbf{W}}$ $Q_{K}^{\mathbf{W}}$=...=$U_{k_1}(\hat{\mathbf{W}})$+$2^{k_1}Q_{k_2}(\hat{\mathbf{W}})$. This structure implies that both the low-order unsigned component $U_{k_1}(\hat{\mathbf{W}})$ and the high-order signed component $Q_{k_2}(\hat{\mathbf{W}})$ are derived using the \emph{same} scaling factor $s_{K}^{\mathbf{W}}$ (from the K-bit quantization). This seems counterintuitive, and it's not clear how $U_{k_1}(\hat{\mathbf{W}})$ is defined. This ambiguity obscures the precise mathematical definition of the decomposition.**
>
> Thank you for raising this point about the notation in Equation 6. The scaling factor is the same in the low-order and high-order components.
>
> *1 - Scaling factor from $K$-bit quantization.*
>
> The scale $s_{K}^{\hat{\mathbf{W}}}$ comes directly from the $K$-bit quantization of the smoothed weights:
>
> \begin{equation}
> Q_K(\hat{\mathbf{W}}) = s_{K}^{\hat{\mathbf{W}}} Q_{K}^{\hat{\mathbf{W}}},
> \end{equation}
>
> where $Q_{K}^{\hat{\mathbf{W}}}$ is the $K$-bit integer representation of $\hat{\mathbf{W}}$.
> We then decompose this $K$-bit integer code into a $k_1$-bit low-order unsigned component $U_{k_1}(\hat{\mathbf{W}})$ and a $k_2$-bit high-order signed component $Q_{k_2}(\hat{\mathbf{W}})$:
>
> \begin{equation}
> Q_{K}^{\hat{\mathbf{W}}}= U_{k_1}(\hat{\mathbf{W}}) + 2^{k_1} Q_{k_2}(\hat{\mathbf{W}}).
> \end{equation}
>
> *2-Why a single shared scaling factor is used.*
>
> The factor $2^{k_1}$ in Equation 6 is applied at the integer level to preserve numerical consistency (i.e., it plays the role of a bit shift). It does not introduce any additional scale. Since both $U_{k_1}(\hat{\mathbf{W}})$ and $Q_{k_2}(\hat{\mathbf{W}})$ are obtained from the same $K$-bit quantized representation $Q_{K}^{\hat{\mathbf{W}}}$, it is natural that they share the same scale $s_{K}^{\hat{\mathbf{W}}}$.
>
> >**Question3-The notation for quantized activations in the proof of Theorem~4.3 is confusing. The proof introduces $Q_{k_2}(\hat{\mathbf{X}})$ and even $Q_{k_2}(\hat{\mathbf{X}})$, but the $(k_1,k_2)$ bit-slicing scheme is defined for weights, not activations. This makes the proof difficult to follow and its validity unclear.**
>
> Thank you for pointing this out.
>
> The use of $Q_{k_1}(\hat{\mathbf{X}})$ and $Q_{k_2}(\hat{\mathbf{X}})$ in Theorem 4.3 is a typographical mistake, since the ((k_1, k_2)) bit-slicing scheme is only defined for weights, not activations. We have corrected these notations in the revised version and updated the proof accordingly, without changing the final error bound. The corrected proof has been revised in the manuscript.

---

> > ### Author Response · Authors · 2025-11-26
> > **Author Response to Reviewer QWVN (Part 2)**
> >
> > >**Question4-Could the authors please explain how the $k_2=2$ bits for the CPU are implemented? Specifically, how are the "Highest INT1" and "2nd Highest INT1" (from Figure 3c) combined and processed to match the single $\mathbf{Q}_{k_2}^{\hat{\mathbf{W}}}$ term in Equation 7?**
> >
> > Thanks for your comment.
> >
> > 1 - Implementation of the $k_2=2$ bits on the CPU.
> >
> > In our setting, the total bit-width is fixed to $K = 6$, with $k_1 = 4$ bits allocated to the GPU and $k_2 = 2$ bits to the CPU. At the implementation level (Figure 3(c)), these two high-order bits are stored as two INT1 bit-planes, labeled as "2nd Highest INT1'' and "Highest INT1''. We denote them by $S_1, S_2 \in \{0,1\}$, respectively. On the CPU side, we apply the LUT-based kernels separately to $S_1$ and $S_2$, so that each bit-plane contributes its own lookup-based result before they are combined.
> >
> > 2- How the two INT1 planes match the single $\mathbf{Q}_{k_2}^{\hat{\mathbf{W}}}$ term.
> >
> > The two INT1 planes $S_1$ and $S_2$ are combined using their bit weights:
> > \begin{equation}
> > 2^{k_1}(S_1 + 2 S_2) = 2^{k_1} \mathbf{Q}_{k_2}^{\hat{\mathbf{W}}}
> > \end{equation}
> >
> > This is to say, binary representaiton $S_1 + 2 S_2$ is exactly the 2-bit integer representation $Q_{k_2}^{\hat{\mathbf{W}}}$. In other words, the “Highest INT1” and “2nd Highest INT1” in Figure 3(c) are simply the implementation-level binary expansion of $Q_{k_2}^{\hat{\mathbf{W}}}$, and are numerically equivalent to that theoretical term.
> >
> >
> > >**Question5-Could the authors provide a more concrete explanation or pseudo-code for the "sign restoration" mechanism? How is "subtracting 16" implemented on a UINT4 GPU value, and what is its precise interaction with the "dividing by 2" step?**
> >
> > Thanks for your comment.
> >
> > The goal of the sign restoration step is to make the high-order bit-planes of positive and negative weights exhibit a symmetric numeric pattern for the LUT. Under two's-complement, we apply an equivalent reparameterization for negative weights. For negative weights, we shift one unit of magnitude from the low part to the high part:
> >
> > \begin{equation}
> > U_{k_1}(\hat{\mathbf{W}}) + 2^{k_1} Q_{k_2}^{\hat{\mathbf{W}}}
> > = (U_{k_1}(\hat{\mathbf{W}}) - 2^{k_1}) + 2^{k_1} (Q_{k_2}^{\hat{\mathbf{W}}} + 1).
> > \end{equation}
> >
> > where $k_1 = 4$ in our setting, so this corresponds to subtracting $16$ from the low 4-bit part and incrementing the high-order term $Q_{k_2}^{\hat{\mathbf{W}}}$ by $1$ (in fact $1 * 2^4$ ). This transformation keeps the combined value exactly unchanged, but makes the high-order bits more symmetric for LUT construction.
> >
> > In implementation, the low part $U_{k_1}(\hat{\mathbf{W}})$ on the GPU is stored as a 4-bit unsigned integer $u_{\text{uint4}} \in \{0,\dots,15\}$. For weights that require sign restoration, we first cast this 4-bit value to a regular integer and apply
> >
> > \begin{equation}
> > u_{\text{signed}} = u_{\text{uint4}} - 16,
> > \end{equation}
> >
> > shifting its logical range from $[0,15]$ to $[-16,15]$, which corresponds to an INT5 representation. To continue using standard INT4 kernels, we then divide this signed value by $2$ and absorb the factor into the dequantization scale ($s_{K}^{\hat{\mathbf{W}}}$):
> >
> > \begin{equation}
> > u_{\text{int4}} = \frac{u_{\text{signed}}}{2},
> > \end{equation}
> >
> > As a result, the GPU effectively uses an INT4 value in $[-8,7]$ with a doubled scale, while the CPU side uses the updated high-order term $\mathbf{Q}_{k_2}^{\hat{\mathbf{W}}}$ in the LUT computation. This implementation exactly realizes the above reparameterization, so the numerical result and the theoretical error bound remain unchanged, while the computation on the GPU stays efficient.
> >
> >
> >
> > >**Question6-What was the group size $g$ used for the LUT in the main experiments? Could the authors provide a sensitivity analysis on how $g$ affects end-to-end latency?**
> >
> > Thank you for the question.
> >
> > Our design follows T-MAC [1]: with 8-bit LUT entries, (g = 4) yields ($2^4 = 16$) table entries, which fit exactly into a 128-bit NEON/AVX2 register and can be accessed by a single -table-lookup instruction.
> >
> >  ● If $g$ is much larger, the table can no longer reside fully in registers, and it's unable to leverage NEON/AVX2 instructions on CPU.
> >
> >  ● If $g$ is much smaller, the number of groups increases and the overhead grows faster than the benefit. For these hardware and efficiency reasons, we fix (g = 4) in our implementation and experiments, and will explicitly state this choice in the paper.
> >
> > [1] T-MAC: CPU Renaissance via Table Lookup for Low-Bit LLM Deployment on Edge EuroSys 2025

---

> > > ### Author Response · Authors · 2025-11-26
> > > **Author Response to Reviewer QWVN (Part 3)**
> > >
> > > >**Question7-Does the ZipperQuant pipeline indeed require both quantizing $\hat{X}$ for the GPU and sending the full-precision $\hat{X}$ to the CPU, as implied by Equation 7? If so, what is the measured overhead of this dual requirement?**
> > >
> > > Thank you for the question.
> > >
> > > In fact, the two process of quantization and transmission are parallized. They run at the same time. We added a detailed latency breakdown ($\mu$s) on NVIDIA RTX 4090 for LLaMA3-8B, covering representative linear layers (e.g., q_proj, o_proj, k_proj, v_proj, down_proj) and batch sizes.
> > >
> > > | bz | layer | GPU INT4 GEMM | GPU Quantize | GPU Total | CPU D2H | CPU LUT | CPU H2D | CPU Total |
> > > |---:|:------|--------------:|-------------:|----------:|--------:|--------:|--------:|----------:|
> > > | 1 | q_proj, o_proj | 116 | 78  | 194.00 | 16.0 | 165.0 | 17.0 | 198.1 |
> > > |  | k_proj, v_proj | 112 | 31  | 143.00 | 15.3 | 113.0 | 15.5 | 143.7 |
> > > |  | down_proj      | 170| 123 | 293.00 | 22.6 | 249.0 | 23.3 | 295.0 |
> > > | 2 | q_proj, o_proj | 124 | 103 | 227.00 | 17.1 | 201.3 | 17.5 | 235.9 |
> > > |  | k_proj, v_proj | 114 | 51  | 165.00 | 15.5 | 136.7 | 15.8 | 168.1 |
> > > |  | down_proj      | 269 | 213 | 482.00 | 22.3 | 423.3 | 22.5 | 468.1 |
> > > | 4 | q_proj, o_proj | 173 | 201 | 374.00 | 19.2 | 330.0 | 19.7 | 368.9 |
> > > |  | k_proj, v_proj | 115 | 65  | 180.00 | 16.0 | 149.2 | 16.5 | 181.7 |
> > > |  | down_proj      | 452 | 312 | 764.00 | 30.6 | 522.9 | 30.2 | 583.7 |
> > > | 8 | q_proj, o_proj | 318 | 285 | 603.00 | 24.3 | 528.0 | 23.1 | 575.5 |
> > > |  | k_proj, v_proj | 127 | 93  | 220.00 | 18.1 | 196.9 | 18.8 | 233.8 |
> > > |  | down_proj      | 801 | 531 | 1332.00 | 46.2 | 1045.8 | 46.7 | 1138.6 |
> > >
> > > >**Question8-Could the authors provide a direct performance (latency/throughput) comparison against other W4A4 methods like QuaRot and SpinQuant, even if it requires discussing the impact of their respective overheads?**
> > > Thank you for the question.
> > >
> > > We further conduct end-to-end throughput comparisons with state-of-the-art W4A4 (QuaRot [1], FlatQuant [2]), and W4A16 (AWQ [3]) systems. The results show that ZipperQuant provides a clear efficiency advantage, achieving about 7.4–13.8$\times$ higher throughput than current W4A4 methods.
> > >
> > > | **Method** |   **bz=1**   |   **bz=2**   |   **bz=4**   |   **bz=8**   |
> > > |----------:|----------:|-------------:|----------------:|--------:|
> > > | TensorRT-LLM [4] (W16A16)| 61.18 | 117.1| 228.0 |418.2 |
> > > | AWQ [3] (W4A4) | 55.2|116.5 | 191.8 |506.4 |
> > > | FlatQuant [2] (W4A4)|9.4| 17.9|35.72 |72.66|
> > > | QuaRot [1] (W4A4) |14.9| 27.7| 55.3 |110.3|
> > > | ZipperQuant (W4A4)|124.8 | 247.2 |407.8 | 817.9|
> > >
> > > [1] QuaRot: Outlier-Free 4-Bit Inference in Rotated LLMs, NeurIPS 2024.
> > >
> > > [2] FlatQuant: Flatness Matters for LLM Quantization, ICML 2025.
> > >
> > > [3] AWQ: Activation-aware Weight Quantization for LLM Compression and Acceleration, MLSys 2024.
> > >
> > > [4] TensorRT-LLM: An open-source TensorRT-based LLM inference and optimization library, NVIDIA, 2025.

---

> > > > ### Author Response · Authors · 2025-11-26
> > > > **Author Response to Reviewer QWVN (Part 4)**
> > > >
> > > > >**Question9-Does the low latency of the CPU path, shown for QKV projections in Figure 4a, also hold for the larger FFN layers? Can the authors confirm the CPU offload does not become a bottleneck in any part of the models tested?**
> > > >
> > > > Thanks for your question. Compared with QKV projection, latency in FFN is also suitable. We provide a detailed latency breakdown ($\mu$ s) on NVIDIA RTX 4090 for LLaMA3-8B again to show the result.
> > > >
> > > > | bz | layer | GPU INT4 GEMM | GPU Quantize | GPU Total|CPU D2H|CPU LUT|CPU H2D| CPU Total|
> > > > |---:|:------|--------------:|-------------:|----------:|--------:|--------:|--------:|----------:|
> > > > | 1 | q_proj, o_proj | 116 | 78  | 194.00 | 16.0 | 165.0 | 17.0 | 198.1 |
> > > > |  | k_proj, v_proj | 112 | 31  | 143.00 | 15.3 | 113.0 | 15.5 | 143.7 |
> > > > |  | down_proj      | 170 | 123 | 293.00 | 22.6 | 249.0 | 23.3 | 295.0 |
> > > > | 2 | q_proj, o_proj | 124 | 103 | 227.00 | 17.1 | 201.3 | 17.5 | 235.9 |
> > > > |  | k_proj, v_proj | 114 | 51  | 165.00 | 15.5 | 136.7 | 15.8 | 168.1 |
> > > > |  | down_proj      | 269 | 213 | 482.00 | 22.3 | 423.3 | 22.5 | 468.1 |
> > > > | 4 | q_proj, o_proj | 173 | 201 | 374.00 | 19.2 | 330.0 | 19.7 | 368.9 |
> > > > |  | k_proj, v_proj | 115 | 65  | 180.00 | 16.0 | 149.2 | 16.5 | 181.7 |
> > > > |  | down_proj      | 452 | 312 | 764.00 | 30.6 | 522.9 | 30.2 | 583.7 |
> > > > | 8 | q_proj, o_proj | 318 | 285 | 603.00 | 24.3 | 528.0 | 23.1 | 575.5 |
> > > > |  | k_proj, v_proj | 127 | 93  | 220.00 | 18.1 | 196.9 | 18.8 | 233.8 |
> > > > |  | down_proj      | 801 | 531 | 1332.00 | 46.2 | 1045.8 | 46.7 | 1138.6 |
> > > >
> > > > At the same time, end-to-end experiement also illustrates the robust of our method. We provide end-to-end throughput comparisons against state-of-the-art W4A4 (QuaRot [1], FlatQuant [2]) and W4A16 (AWQ [3]) systems.
> > > >
> > > > | **Method** |   **bz=1**   |   **bz=2**   |   **bz=4**|   **bz=8**   |
> > > > |----------:|----------:|-------------:|----------------:|--------:|
> > > > | TensorRT-LLM [4] (W16A16)| 61.18 | 117.1| 228.0 |418.2 |
> > > > | AWQ [3] (W4A4) | 55.2|116.5 | 191.8 |506.4 |
> > > > | FlatQuant [2] (W4A4)|9.4| 17.9|35.72 |72.66|
> > > > | QuaRot [1] (W4A4) |14.9| 27.7| 55.3|110.3|
> > > > | ZipperQuant (W4A4)|124.8 |247.2|407.8|817.9|
> > > >
> > > > [1] QuaRot: Outlier-Free 4-Bit Inference in Rotated LLMs, NeurIPS 2024.
> > > >
> > > > [2] FlatQuant: Flatness Matters for LLM Quantization, ICML 2025.
> > > >
> > > > [3] AWQ: Activation-aware Weight Quantization for LLM Compression and Acceleration, MLSys 2024.
> > > >
> > > > [4] TensorRT-LLM: An open-source TensorRT-based LLM inference and optimization library, NVIDIA, 2025.
> > > >
> > > > >**Question10-How robust is the method's performance to the high-order bit sparsity assumption? Does this high sparsity (Figure 3b) hold for FFN layers as well? What is the performance impact if sparsity decreases?**
> > > >
> > > > Thanks for your question.
> > > >
> > > > *1 - Does high sparsity hold for FFN layers as well?*
> > > >
> > > > To verify that the high-order bit sparsity is not specific to attention weights, we further measure the sparsity of the 1st and 2nd highest bit-planes on all FFN weight matrices (down_proj, gate_proj, up_proj) of Qwen3-4B. The new table below shows that the higher-bit planes remain highly sparse across all 34 layers: the 1st highest bit consistently exceeds 99.7% sparsity, and the 2nd highest bit stays around 90%-92% for all three FFN projections. Moreover, we can systematically control the sparsity level of the high-order bits by adjusting the smoothing factor.
> > > >
> > > > | layer ID | bit | layer_0 | layer_3 | layer_6 | layer_9 | layer_12 | layer_15 | layer_18 | layer_21 | layer_24 | layer_27 | layer_30 | layer_33 |
> > > > |---|---|---:|---:|---:|---:|---:|---:|---:|---:|---:|---:|---:|---:|
> > > > | down_proj | 1st | 99.84 | 99.65 | 99.82 | 99.79 | 99.79 | 99.75 | 99.74 | 99.74 | 99.8 | 99.83 | 99.87 | 99.86 |
> > > > |         | 2nd | 90.42 | 90.6 | 90.49 | 90.52 | 90.48 | 90.6 | 90.6 | 90.55 | 90.57 | 90.54 | 90.43 | 90.41 |
> > > > | gate_proj | 1st | 99.86 | 99.67 | 99.84 | 99.84 | 99.81 | 99.8 | 99.77 | 99.76 | 99.8 | 99.84 | 99.87 | 99.84 |
> > > > |        | 2nd | 90.34 | 92.36 | 90.37 | 90.37 | 90.39 | 90.4 | 90.45 | 90.45 | 90.39 | 90.37 | 90.35 | 90.36 |
> > > > | up_proj | 1st | 99.87 | 99.76 | 99.85 | 99.84 | 99.81 | 99.81 | 99.79 | 99.8 | 99.84 | 99.86 | 99.88 | 99.87 |
> > > > |       | 2nd | 90.32 | 90.41 | 90.35 | 90.35 | 90.41 | 90.4 | 90.43 | 90.38 | 90.36 | 90.35 | 90.32 | 90.33 |
> > > >
> > > > *2-Performance impact when sparsity decreases.*
> > > > We also conduct a controlled study where we vary the enforced sparsity level of the high-order bit-planes in FFN layers and measure the corresponding per-layer LUT time (\mu s) in CPU. The results show the expected trend: as sparsity increased from 90% to 99.5%, the per-layer LUT cost also decreases.
> > > > | layer ID | 99.5% | 98% | 95% | 93% | 90%|
> > > > |---|---|---:|---:|---:|---:|
> > > > | down_proj | 90.9 | 135.3 | 160.4 | 210.3 | 249.0 |
> > > > | gate_proj | 100.5 | 141.4 | 168.7 | 214.6 | 255.0 |
> > > > | up_proj | 100.8 | 141.8 | 167.9 |213.9|254.2|

---

> > > > > ### Author Response · Authors · 2025-11-26
> > > > > **Author Response to Reviewer QWVN (Part 5)**
> > > > >
> > > > > >**Question11-Do the authors have any insights or preliminary data on how this GPU-CPU disaggregation strategy might perform on different hardware architectures, such as server-grade GPUs or edge devices?**
> > > > >
> > > > > Thanks for your question.ZipperQuant demonstrates consistent robustness across heterogeneous GPU–CPU platforms. We evaluate LLaMA3-8B with ZipperQuant on a **workstation** configured with an NVIDIA RTX4090 (24GB) GPU, an Intel Xeon(R) Gold 6430 CPU, and 120 GB of DDR5 host memory. To further assess robustness beyond the workstation setting, we also test ZipperQuant on an **AI inference server** equipped with an NVIDIA L20 (48GB) GPU, an Intel Xeon(R) Platinum 8457C CPU, and 100GB of DDR5 host memory.
> > > > >
> > > > > |**Method**|**bz=1**|**bz=2**|**bz=4**|**bz=8**|
> > > > > |----:|------:|-----:|-----:|-----:|
> > > > > | TensorRT-LLM [1] (W16A16)|47.8|95.3|190.4|375.9|
> > > > > | FlatQuant [2] (W4A4)|18.8|33.57|63.5|143.39|
> > > > > | QuaRot [3] (W4A4)|11.83|21.68|40.52|192.45|
> > > > > | **ZipperQuant (W4A4)**|**81.62**|**141.86**|**237.37**|**622.93**|
> > > > >
> > > > > [1] TensorRT-LLM: An open-source TensorRT-based LLM inference and optimization library, NVIDIA, 2025.
> > > > >
> > > > > [2] FlatQuant: Flatness Matters for LLM Quantization, ICML 2025.
> > > > >
> > > > > [3] QuaRot: Outlier-Free 4-Bit Inference in Rotated LLMs, NeurIPS 2024.
> > > > >
> > > > >
> > > > > We warmly welcome any additional comments or suggestions you may have. If our responses have addressed your concerns, could you please consider upgrading the score. Thank you very much!
> > > > >
> > > > > Best regards,
> > > > >
> > > > > The Authors

---

### Official Review · Reviewer_kTRi · 2025-11-01

**Soundness:** 3
**Presentation:** 3
**Contribution:** 3
**Rating:** 6
**Confidence:** 3

**Summary:**

This paper introduces a new quantization method to achieve accurate, efficient LLM inference via finer-grained bit-level disaggregation across GPU and CPU. A LUT-based approach is further proposed to accelerate the high-order bit computation on CPU. Experimental results show the effectiveness of the proposed method.

**Strengths:**

+ finer-grained bit-level disaggregation across GPU and CPU to achieve accurate, efficient LLM inference
+ LUT-based approach is further proposed to accelerate the high-order bit computation on CPU

**Weaknesses:**

- accuracy comparison seems unfair as ZipperQuant does 6-bit quantization rather than 4-bit one
- no direct comparison with existing disaggregation approaches

**Questions:**

It looks ZipperQuant does 6-bit quantization. Is it fair to compare with 4-bit quantization for accuracy? For latency, how does ZipperQuant compare with existing disaggregation-based approaches?

---

> ### Author Response · Authors · 2025-11-24
> **Author Response to Reviewer kTRi**
>
> >**It looks ZipperQuant does 6-bit quantization. Is it fair to compare with 4-bit quantization for accuracy? For latency, how does ZipperQuant compare with existing disaggregation-based approaches?**
>
> *1 - Fair W4A4 Comparison Under the Same GPU Budget.*
>
> Thank you for the question. We would like to clarify a possible misunderstanding about bit-width. ZipperQuant’s GPU path is strictly W4A4: all GPU weights, activations, memory traffic, and GEMMs are 4-bit. Only a sparse set of high-order weight bits is handled on the CPU as a separate correction branch, which does not increase GPU bit-width or footprint. Therefore, our accuracy comparison is fair under the same W4A4 GPU budget. Moreover, prior SOTA methods (e.g., QuaRot [1], SpinQuant [2], FlatQuant [3]) have established W4A4 as a strong accuracy–efficiency point when paired with smoothing and rotations; we follow this setting, where smoothing shifts activation outliers into weights to make A4 quantization stable.
>
> *2 - Comparsion with existing disaggregation-based approaches.*
>
> To the best of our knowledge, we are the first to propose bit-level disaggregation: we leverage the CPU to compute sparse high-bit corrections while keeping the GPU path strictly W4A4, thus avoiding extra GPU burden and heavy data transfers. As shown in the table below, ZipperQuant achieves 1.79-2.11$\times$ higher throughput than full-precision TensorRT-LLM engine, and delivers a further 7.37–8.92$\times$ speedup compared with state-of-the-art W4A4 systems.
> | **Method** |   **bz=1**   |   **bz=2**   |   **bz=4**   |   **bz=8**   |
> |----------:|----------:|-------------:|----------------:|--------:|
> | TensorRT-LLM [4] (W16A16）|   61.18   |   117.1   |   228.0   |   418.2    |
> | FlatQuant [3] (W4A4) | 9.4 | 17.9 | 35.72 | 72.66 |
> | QuaRot [1] (W4A4) | 14.9 | 27.7 | 55.3 | 110.3 |
> | **ZipperQuant (W4A4)** |  **124.8**  |  **247.2**  |  **407.8**   |  **817.9**  |
>
> [1] QuaRot: Outlier-Free 4-Bit Inference in Rotated LLMs, NeurIPS 2024.
>
> [2] SpinQuant: LLM quantization with learned rotations, ICLR 2025.
>
> [3] FlatQuant: Flatness Matters for LLM Quantization, ICML 2025.
>
> [4] TensorRT-LLM: An open-source TensorRT-based LLM inference and optimization library, NVIDIA, 2025.
>
> We warmly welcome any additional comments or suggestions you may have. If our responses have addressed your concerns, could you please consider upgrading the score. Thank you very much!
>
> Best regards,
>
> The Authors

---

### Official Review · Reviewer_PfpY · 2025-11-01

**Soundness:** 3
**Presentation:** 3
**Contribution:** 3
**Rating:** 4
**Confidence:** 4

**Summary:**

The paper proposes ZipperQuant, a post-training 4-bit quantization and execution scheme that splits work across GPU and CPU. After a smoothing step that “absorbs” activation outliers into the weights, each weight tensor is decomposed into low-order and high-order bit components. The GPU executes inliers and the low-order bits with INT4 kernels, while only the sparse high-order bits are offloaded to the CPU and applied at higher precision. To reduce CPU cost, the authors precompute a lookup-table (LUT) over the small set of high-order bit patterns and replace multiplications with table lookups plus accumulation. Experiments on an RTX 4090 claim near-FP16 accuracy and speedups over a W4A16 baseline.

**Strengths:**

1. Clear systems idea: The bit-level disaggregation is an interesting twist on prior value-based outlier handling, with a plausible path to better overlap and lower transfer volume.

2. Hybrid deployment perspective: Embraces real-world heterogeneity (CPU+GPU) rather than assuming an all-GPU pipeline; the paper surfaces important engineering considerations (transfer, overlap, cache effects).

**Weaknesses:**

1.  The core contribution is mainly a systems design/implementation choice (bit-wise split + LUT), while the “smoothing” component appears conceptually close to prior activation-aware methods (e.g., SmoothQuant-style weight–activation rebalancing). The paper does not articulate a clearly new quantization algorithm beyond the execution strategy.

2. The headline comparison is against a W4A16 baseline, which may be weak. It’s unclear how ZipperQuant fares against strong end-to-end stacks with optimized INT4 paths and outlier handling (e.g., TensorRT-LLM, vLLM with optimized kernels, AWQ/OmniQuant/ZeroQuant variants).

3. The claimed gains are reported on RTX 4090; the approach’s benefit likely depends on GPU:CPU FLOP ratio, memory bandwidth, cache sizes, and interconnect (PCIe 4/5, NVLink). There is no systematic study showing that the speed-accuracy trade-off is robust across different CPU classes (e.g., AVX2 vs AVX-512/AMX), GPUs, or buses.

**Questions:**

How does ZipperQuant’s end-to-end throughput/latency compare with TensorRT-LLM, highly tuned vLLM INT4 kernels, and recent activation-aware 4-bit baselines (e.g., AWQ, OmniQuant, SmoothQuant)? Please include same hardware, same batch/seq settings, and decode vs prefill regimes.

---

> ### Author Response · Authors · 2025-11-24
> **Author Response to Reviewer PfpY (Part 1)**
>
> >**Weakness 1 - The core contribution is mainly a systems design/implementation choice (bit-wise split + LUT), while the “smoothing” component appears conceptually close to prior activation-aware methods (e.g., SmoothQuant-style weight–activation rebalancing). The paper does not articulate a clearly new quantization algorithm beyond the execution strategy.**
>
> Thank you for your comment. We respectfully emphasize that ZipperQuant is not merely an execution or systems optimization, but introduces a new quantization formulation that cannot be obtained by prior methods.
>
> *1 - Novel quantization representation.*
>
> ZipperQuant proposes a bit-plane factorization of the quantized weight (Eq. (6) in the paper):
> \begin{equation}
>     Q_K(\hat{\mathbf{W}}) = U_{k_1}(\hat{\mathbf{W}}) + 2^{k_1}Q_{k_2}(\hat{\mathbf{W}}),
> \end{equation}
> which separates the representation into low-order and high-order components with different numerical roles. This representation changes the quantization operator itself, unlike value-based methods (SVDQuant [1], DecDEC [2]) that operate on the quantized tensor as a whole.
>
> *2 - New theoretical properties.*
>
> In Section 4.2, we derive a new quantization error bound (Theorem 4.3) that is strictly tighter than standard k-bit quantization bounds. This theoretical result is unique to our bit-plane decomposition and does not arise in prior activation-aware or rotation-based methods.
>
> *3 - Algorithmic mechanism, not a system trick.*
>
> The Zipper Engine (Section 4.3)—including bit-plane LUTs, sign restoration, and unified sparsity—is algorithmically required to make the above quantization representation feasible.
> Without bit-plane disaggregation, performing W4 quantization while preserving W6-level representational range is impossible.
>
> *4 - Independence from smoothing.*
>
> While smoothing resembles prior activation-aware methods, it is not the core contribution. It only enables A4 quantization. The key novelty is that ZipperQuant redefines how higher-bit expressivity can be achieved under a 4-bit footprint through bit-level decomposition—a direction not explored in previous PTQ literature.
>
> For these reasons, ZipperQuant proposes a new quantization paradigm, with system co-design playing a supporting—not primary—role.
>
> [1] SVDQuant: Absorbing Outliers by Low-Rank Components for 4-Bit Diffusion Models, ICLR 2025 Spotlight
>
> [2] DecDEC: A Systems Approach to Advancing Low-Bit LLM Quantization, OSDI 2025
>
> >**Weakness 2 - The headline comparison is against a W4A16 baseline, which may be weak. It’s unclear how ZipperQuant fares against strong end-to-end stacks with optimized INT4 paths and outlier handling (e.g., TensorRT-LLM, vLLM with optimized kernels, AWQ/OmniQuant/ZeroQuant variants).**
>
> Thanks for your valuable question.
> We further compare ZipperQuant under the W4A4 setting on LLaMA3-8B against several state-of-the-art post-training quantization methods, including FlatQuant, and QuaRot. We report quantization time, accuracy on six zero-shot tasks, and end-to-end throughput, providing a more comprehensive view of how ZipperQuant performs relative to strong INT4 baselines with outlier handling.
> | **Method** |   **bz=1**   |   **bz=2**   |   **bz=4**   |   **bz=8**   |
> |----------:|----------:|-------------:|----------------:|--------:|
> | TensorRT-LLM [1] (W16A16)|   61.18   |   117.1   |   228.0   |   418.2    |
> | FlatQuant [2] (W4A4)| 9.4 | 17.9 | 35.72 | 72.66 |
> | QuaRot [3] (W4A4)| 14.9 | 27.7 | 55.3 | 110.3 |
> | **ZipperQuant (W4A4)**|  **124.8**  |  **247.2**  |  **407.8**   |  **817.9**  |
>
> [1] TensorRT-LLM: An open-source TensorRT-based LLM inference and optimization library, NVIDIA, 2025.
>
> [2] FlatQuant: Flatness Matters for LLM Quantization, ICML 2025.
>
> [3] QuaRot: Outlier-Free 4-Bit Inference in Rotated LLMs, NeurIPS 2024.

---

> > ### Author Response · Authors · 2025-11-24
> > **Author Response to Reviewer PfpY (Part 2)**
> >
> > >**Weakness 3 - The claimed gains are reported on RTX 4090; the approach’s benefit likely depends on GPU:CPU FLOP ratio, memory bandwidth, cache sizes, and interconnect (PCIe 4/5, NVLink). There is no systematic study showing that the speed-accuracy trade-off is robust across different CPU classes (e.g., AVX2 vs AVX-512/AMX), GPUs, or buses.**
> >
> > ZipperQuant is robust across different GPU and CPU models. We have evaluated ZipperQuant for LLaMA3-8B **Workstation**: powered by an NVIDIA RTX 4090 (24 GB) GPU, Intel Xeon(R) Gold 6430 CPU, 120 GB DDR5 host memory. To examine hardware robustness beyond workstation, we additionally evaluate ZipperQuant on **AI inference server**: equipped with an NVIDIA L20 (48 GB) GPU, Intel Xeon(R) Platinum 8457C CPU, 100 GB DDR5 host memory.
> > | **Method** |   **bz=1**   |   **bz=2**   |   **bz=4**   |   **bz=8**   |
> > |----------:|--------------------:|---------------------:|----------------------:|-----------------------:|
> > | TensorRT-LLM [1] (W16A16)|   47.8   |   95.3   |   190.4   |   375.9    |
> > | FlatQuant [2] (W4A4)|     18.8    |    33.57    |    63.5    |   143.39    |
> > | QuaRot [3] (W4A4)|     11.83    |    21.68    |    40.52    |   192.45    |
> > | **ZipperQuant (W4A4)**|  **81.62**  |  **141.86**  |  **237.37**  |  **622.93**  |
> >
> > Across batch sizes 1 to 8, ZipperQuant remains the fastest W4A4 method, delivering roughly 3–7$\times$ higher throughput than FlatQuant and QuaRot, and showing the same favorable trend versus FP16 and the AWQ baseline as on RTX 4090, which indicates that our speed–accuracy trade-off is stable on a different GPU and CPU platform. Our CPU-side Zipper engine is implemented with Intel AVX2 intrinsics, so it applies to any Intel CPU that supports AVX2.
> >
> > [1] TensorRT-LLM: An open-source TensorRT-based LLM inference and optimization library, NVIDIA, 2025.
> >
> > [2] FlatQuant: Flatness Matters for LLM Quantization, ICML 2025.
> >
> > [3] QuaRot: Outlier-Free 4-Bit Inference in Rotated LLMs, NeurIPS 2024.
> >
> > We warmly welcome any additional comments or suggestions you may have. If our responses have addressed your concerns, could you please consider upgrading the score. Thank you very much!
> >
> > Best regards,
> >
> > The Authors

---

### Official Review · Reviewer_xBns · 2025-11-03

**Soundness:** 2
**Presentation:** 2
**Contribution:** 2
**Rating:** 4
**Confidence:** 3

**Summary:**

This paper proposes ZipperQuant, a hybrid GPU–CPU execution scheme for 4-bit PTQ LLM inference.
The method splits  modelsweights per bit-plane after smoothing, low-order bits → GPU int4 path (dense, fast) while sparse high-order bits → CPU LUT path (lookup w/o multiply)
so the “inlier” part of outliers stays on GPU, and only the sparse “heavy” bits get offloaded to CPU.

**Strengths:**

The paper addresses a real deployment bottleneck by leveraging the sparsity of high-order bit planes.

The proposed method demonstrates improvements over baselines

**Weaknesses:**

1. Wrong reference year: Some method has wrong year, like AWQ is a 2023 method
2. Weak baselines: The paper seems to compare with the result of SpinQuant without Hadamard matrix, which is much weaker than their version using Hadamard. Also, there are stronger baselines like FlatQuant that need to be compared. Additionally, why do the author compare the speed up and memory with only AWQ, which is kinda old?

3. Technical novelty: The novelty feel weak to me, as they seems to be engineering trick to improve efficiency, but the results actually show improvement, so what is the key behind the performance improvement over spinQuant?

**Questions:**

please see weakness

---

> ### Author Response · Authors · 2025-11-24
> **Author Response to Reviewer xBns (Part 1)**
>
> >**Weakness 1 - Wrong reference year: Some method has wrong year, like AWQ is a 2023 method**
>
> Thank you for your question. AWQ is officially published at MLSys 2024, so we cite the conference paper instead of the 2023 arXiv preprint. We have also carefully re-checked all references.
>
> >**Weakness 2 - Weak baselines: The paper seems to compare with the result of SpinQuant without Hadamard matrix, which is much weaker than their version using Hadamard. Also, there are stronger baselines like FlatQuant that need to be compared. Additionally, why do the author compare the speed up and memory with only AWQ, which is kind a old?**
>
> Thank you for your question.
>
> *1 - Fair comparison under the same online-overhead budget*
>
> Our target is W4A4 inference with almost zero extra online overhead. ZipperQuant requires no retraining and adds no online rotation cost, whereas SpinQuant with Hadamard introduces additional online rotation. Moreover, our method is orthogonal to rotation: even on top of SpinQuant without Hadamard, Zipper still brings clear accuracy gains (Table 2), showing the improvement comes from bit-level outlier correction.
>
> *2 - Comparisons against stronger W4A4 systems*
>
> We have added end-to-end throughput and accuracy comparisons (on six zero-shot tasks) for LLaMA3-8B against strong and recent W4A4 systems, including OmniQuant, FlatQuant, and QuaRot, on an NVIDIA RTX 4090. The results show that ZipperQuant delivers about 7.4–13.8$\times$ higher throughput than existing W4A4 systems, while achieving 0.11–16.77 points higher average accuracy across the six tasks. This substantial speedup mainly comes from two factors: (1) unlike prior W4A4 methods, we introduce no extra online computation overhead such as rotations or transformations; and (2) we reimplement and heavily optimize the 4-bit compute kernels in CUDA.
> | **Method** |   **bz=1**   |   **bz=2**   |   **bz=4**   |   **bz=8**   |
> |----------:|----------:|-------------:|----------------:|--------:|
> | TensorRT-LLM [1] (W16A16)|   61.18   |   117.1   |   228.0   |   418.2    |
> | AWQ [2] (W4A16)|     55.22    |    116.53    |    191.85    |   506.45   |
> | FlatQuant [3] (W4A4)| 9.4 | 17.9 | 35.72 | 72.66 |
> | QuaRot [4] (W4A4)| 14.9 | 27.7 | 55.3 | 110.3 |
> | **ZipperQuant (W4A4)**|  **124.8**  | **247.2**  |  **407.8**   | **817.9**  |
>
> |  **Method**  | **ARC_E** | **ARC_C** | **Hellaswag** | **LAMBADA** | **PIQA** | **Winogrande** | **Avg.** |
> |---------------:|----------:|-----------:|-------------:|----------------:|--------:|---------------:|---------:|
> | FlatQuant [3] (W4A4) |  **79.08**  |  52.05  |  76.58 |  70.85 |  79.05 |  72.3  | 71.65 |
> | QuaRot [4] (W4A4) |  62.0  |  41.6  |  64.9 |  67.9 |  75.1 |  62.3  | 60.63 |
> | OmniQuant [5] (W4A4) |  65.95  |  37.71  | 63.26 | 28.84 | 72.52 |  61.64 | 54.99 |
> | **ZipperQuant (W4A4)** | 77.69 |  **52.65**  |  **77.45**  |  **70.91** |  **79.54** | **72.3**  | **71.76** |
>
> *QuaRot's data from L853 in the original manuscript.*
>
> [1] TensorRT-LLM: An open-source TensorRT-based LLM inference and optimization library, NVIDIA, 2025.
>
> [2] AWQ: Activation-aware Weight Quantization for LLM Compression and Acceleration, MLSys 2024.
>
> [3] FlatQuant: Flatness Matters for LLM Quantization, ICML 2025.
>
> [4] QuaRot: Outlier-Free 4-Bit Inference in Rotated LLMs, NeurIPS 2024.
>
> [5] OmniQuant: Omnidirectionally Calibrated Quantization for Large Language Models, ICLR 2024 Spotlight.

---

> > ### Author Response · Authors · 2025-11-24
> > **Author Response to Reviewer xBns (Part 2)**
> >
> > >**Weakness 3 - Technical novelty: The novelty feel weak to me, as they seems to be engineering trick to improve efficiency, but the results actually show improvement, so what is the key behind the performance improvement over spinQuant?
> > Thank you for your comment. We would like to clarify that the gains come from a new quantization formulation and its algorithmic consequences, not from a engineering trick.**
> >
> > Thank you for your valuable question.
> >
> > *1 - Conceptual novelty: a new quantization operator.*
> >
> > ZipperQuant proposes a bit-plane factorization of the quantized weight (Eq. (6) in the paper):
> > \begin{equation}
> >     Q_K(\hat{\mathbf{W}}) = U_{k_1}(\hat{\mathbf{W}}) + 2^{k_1}Q_{k_2}(\hat{\mathbf{W}}),
> > \end{equation}
> > which separates the representation into low-order and high-order components with different numerical roles. This representation changes the quantization operator itself, unlike value-based methods (SVDQuant [1], DecDEC [2]) that operate on the quantized tensor as a whole.
> >
> > *2 - Theoretical novelty: tighter error characterization from bit-plane decomposition.*
> >
> > In Section 4.2, we derive a new quantization error bound (Theorem 4.3) that is strictly tighter than standard k-bit quantization bounds. This theoretical result is unique to our bit-plane decomposition and does not arise in prior activation-aware or rotation-based methods.
> >
> > *3 - Algorithmic novelty enabling the formulation.*
> >
> > The Zipper Engine (Section 4.3)—including bit-plane LUTs, sign restoration, and unified sparsity—is algorithmically required to make the above quantization representation feasible.
> > Without bit-plane disaggregation, performing W4 quantization while preserving W6-level representational range is impossible.
> >
> > *4 - Key reason for improvement over SpinQuant.*
> >
> > SpinQuant makes weights more quantization-friendly via learned rotations, reducing average W4 error, but W4A4 performance can still be dominated by a small residue of large-magnitude outliers that inflate scales after rotation. ZipperQuant directly targets this remaining bottleneck by isolating those outliers into the high-order bit-plane and correcting them sparsely. Our Spin+Zipper ablation (Table 2) confirms the gains are complementary: rotation reduces the bulk error, while bit-plane correction removes the residual outlier-driven error, leading to consistent improvement beyond SpinQuant.
> >
> > [1] SVDQuant: Absorbing Outliers by Low-Rank Components for 4-Bit Diffusion Models, ICLR 2025 Spotlight.
> >
> > [2] DecDEC: A Systems Approach to Advancing Low-Bit LLM Quantization, OSDI 2025.
> >
> > We warmly welcome any additional comments or suggestions you may have. If our responses have addressed your concerns, could you please consider upgrading the score. Thank you very much!
> >
> > Best regards,
> >
> > The Authors

---

### Official Review · Reviewer_ehPS · 2025-11-04

**Soundness:** 3
**Presentation:** 3
**Contribution:** 3
**Rating:** 4
**Confidence:** 4

**Summary:**

This paper proposes a method to represent the larger bit-width quantization with a smaller bit-width dense tensor container the lower-bits and a sparse tensor containing the higher-bits (i.e., 4+2 in its experiments). It illustrates that larger bit-width is important for quantization with a set of equations. The higher-bits tensor is computed on the GPU, and the lower-bit sparse tensor is computed on the CPU with LUT method (with necessary D2H communication of the activation) and then transferred back to GPU for reduction. It shows better accuracy with its W4A4 (actually should be W6A4?) method than the rotation-based works, and better system efficiency than FP16 and AWQ baseline.

**Strengths:**

1. It focuses on the important problem of LLM serving, the post-training quantization.
2. Splitting the tensor into a dense one and a sparse one at bit-level seems new.

**Weaknesses:**

1. The decision of using 6-bits for weight and 4-bits for activation is not well discussed/evaluated. The advantage of "4-bit on GPU and 2-bit on CPU" than "the full 6-bits on GPU" is not clear (e.g., Blackwell supports the 6-bits datatype).
2. Lacks the comparison of some recent works.
3. The system overhead is not clear.
4. Some statement is not accurate.

**Questions:**

1. Why use larger bits for weight and smaller bits for activation, i.e., 4+2 for weight and 4 for activation? Given that LLM inference is easily bound by the memory access of large weight tensor (QServe [1]), the state-of-the-art solution is to use smaller weights and slightly larger activation as for the bit-width. Even with smoothing, the loss coming from the 4-bit activation can be too large.
2. Given that this paper uses 6-bits for the weight quantization, 4-bit on the GPU and 2-bits on the CPU, why not using the Blackwell FP6 datatype directly? It only saves 1/3 GPU memory of using 4-bits when compared to the full 6-bits. The current model serving solutions are not bound by this portion of GPU memory (especially for the small models evaluated in this paper). Besides, the complexity of 4+2 has introduced a lot of CPU and communication overhead.
3. It could be better to compare this work with more related works (e.g., QServe[1] and more mixed-precision quantization works). Besides, it claims W4, but it is actually W6 in the evaluation. This can be misleading.
4. As for the system efficiency, it should have a comparison with the state-of-the-art W4A4 and W4A8 systems, rather than the relatively old system of AWQ and the full precision.
5. The system overhead is not studied. What is the overhead of the H2D communication of the activation and D2H communication of the partial results? How long is the W4A4 GPU kernel and how long is the end-2-end W6A4 execution including the CPU and communication part? Without this breakdown, it is hard to believe this system is practical due to the potential overhead.
6. The equations in section 4.1 are good! But they seem only used for the illustration that the bit-width is important. This is not a problem though.
7. The statement "Q_X and Q_W typically adopt the same bit-width" in L157 is not accurate, e.g., QServe uses W4A8.
8. It would be good to add space between the columns of Table 1. It is hard to differ different columns for its row-1.
9. It seems have not present the ratio of higher bits and lower bits in the evaluated models and tasks. What is the ratio of the higher bits in each portion of the weight?

[1] QServe: W4A8KV4 Quantization and System Co-design for Efficient LLM Serving

---

> ### Author Response · Authors · 2025-11-24
> **Author Response to Reviewer ehPS (Part 1)**
>
> >**Question1 - Why use larger bits for weight and smaller bits for activation, i.e., 4+2 for weight and 4 for activation? Given that LLM inference is easily bound by the memory access of large weight tensor (QServe), the state-of-the-art solution is to use smaller weights and slightly larger activation as for the bit-width. Even with smoothing, the loss coming from the 4-bit activation can be too large.**
>
> Thank you for the valuable question.
>
> *1 - We are not 4+2 for weight and 4 for activation.*
>
> We would like to clarify a possible misunderstanding of our bit-width design. ZipperQuant’s GPU path is strictly W4A4, while the CPU only processes the high-order weight bits, effectively executing a W2A16 branch for the sparse outlier. Thus, the design is not “4+2-bit weights with 4-bit activations” and we do not use larger bits for weight and smaller bits for activation. Our novelty lies in how we then treat the outlier-heavy weights after smoothing. Instead of increasing the GPU bit-width, we bit-disaggregate them: low-order bits stay on the GPU for INT4 GEMM, while only the rare high-order bits are offloaded to the CPU. This reduces weight-related memory traffic and preserves compute intensity.
>
> *2 - 4-bit activation has been a validated accuracy–efficiency point.*
>
> Recent state-of-the-art works such as QuaRot [1], SpinQuant [2], and FlatQuant [3] have shown that 4-bit activation can deliver strong accuracy when paired with smoothing or rotations. We follow the same W4A4 regime: smoothing shifts activation outliers into the weights, which stabilizes A4 quantization and makes W4A4 a fair and strong operating point for comparison. Moreover, native 4-bit compute support is becoming increasingly available on modern hardware (e.g., INT4 Tensor Core accelerators), making W4A4 not only accurate but also a practical and forward-looking deployment choice.
>
> *3 - Result from the existing experiment shows that the loss is acceptable.*
>
> The above conclusion is supported by our experiment results in Table 4: with the same A4 activations, ZipperQuant (W4A4) achieves 71.76 average accuracy, about 0.21 higher than QServe (W4A8) at 71.55, and clearly outperforms other W4A4 baselines, indicating that  activation quantization is no longer the dominant accuracy bottleneck.
> |**Method**|**ARC_E**|**ARC_C**|**Hellaswag**|**LAMBADA**|**PIQA**|**Winogrande**|**Avg.**|
> |-------------:|----------:|-----------:|-------------:|----------------:|--------:|---------------:|---------:|
> | QServe (W4A8) |76.22|52.13|77.27|71.36|79.92|72.38|71.55|
> | OmniQuant (W4A4) |65.95|37.71|63.26|28.84| 72.52 |61.64|54.99|
> | SpinQuant (W4A4) |66.5|45.9|75.9|67.9|68.0|68.5|65.5|
> | **ZipperQuant (W4A4)** |**77.69**|**52.65**|**77.45**|**70.91**|**79.54**|**72.3**|**71.76**|
>
> [1] QuaRot: Outlier-Free 4-Bit Inference in Rotated LLMs, NeurIPS 2024.
>
> [2] SpinQuant: LLM quantization with learned rotations, ICLR 2025.
>
> [3] FlatQuant: Flatness Matters for LLM Quantization, ICML 2025.

---

> > ### Author Response · Authors · 2025-11-24
> > **Author Response to Reviewer ehPS (Part 2)**
> >
> > >**Question 2 - Given that this paper uses 6-bits for the weight quantization, 4-bit on the GPU and 2-bits on the CPU, why not using the Blackwell FP6 datatype directly? It only saves 1/3 GPU memory of using 4-bits when compared to the full 6-bits. The current model serving solutions are not bound by this portion of GPU memory (especially for the small models evaluated in this paper). Besides, the complexity of 4+2 has introduced a lot of CPU and communication overhead.**
> >
> > Thank you for the valuable question. Our choice of 4 bits on the GPU is driven by both compute throughput and memory efficiency.
> >
> > *1 - GPU throughput considerations.*
> >
> > On NVIDIA Blackwell GPUs, FP6 provides throughput close to FP8, whereas FP4/INT4 is ~2$\times$ faster [1]. Switching the GPU path from INT4/FP4 to FP6 would substantially reduce the compute ceiling that ZipperQuant relies on. Our design maintains pure W4A4 GPU execution, ensuring maximum tensor-core utilization.
> >
> > *2 - ZipperQuant is not 6-bit quantization; it is a 4-bit quantization with an auxiliary sparse correction.*
> >
> > Using FP6 would require storing and operating on all weights in 6 bits on the GPU. In contrast, ZipperQuant's GPU path always stores 4-bit weights, with the CPU branch computing only the high-order bit-plane of sparse outliers. This design achieves a property that FP6 cannot provide:
> >
> >  ● FP6: wider representation everywhere → higher memory + lower throughput
> >
> >  ● ZipperQuant: FP4-level memory + FP4-level throughput on GPU + selective high-order correction on CPU → W4 footprint and speed, while recovering W6 accuracy
> >
> > Even if the hardware supports FP6, using it would violate our design goal: maximize GPU efficiency while enabling higher precision only where necessary.
> >
> > *3 - CPU overhead is minimal and overlapped.*
> >
> > The CPU handles only the high-order bit-plane of sparse outliers, and we transfer only the corresponding activation slices. As shown in Section 4.3 and Figure 4(a), this communication is small and fully overlappable with GPU GEMM execution, so it does not bottleneck performance.
> >
> > [1] NVIDIA RTX BLACKWELL GPU ARCHITECTURE.
> >
> > >**Question3 - It could be better to compare this work with more related works (e.g., QServe[1] and more mixed-precision quantization works). Besides, it claims W4, but it is actually W6 in the evaluation. This can be misleading.**
> >
> > Thank you for the valuable suggestion.
> >
> > *1-Comparsion with mixed-precision baseline.*
> >
> > Following your suggestion, we add QServe [1] under its default W4A8 setting and report accuracy on six zero-shot tasks. The results show that ZipperQuant, despite operating at W4A4 precision, achieves 0.21 points higher average accuracy than QServe’s W4A8, and outperforms state-of-the-art W4A4 systems by 6.2–16.7 points.
> > |  **Method**  | **ARC_E** | **ARC_C** | **Hellaswag** | **LAMBADA** | **PIQA** | **Winogrande** | **Avg.** |
> > |------------:|----------:|-----------:|-------------:|----------------:|--------:|---------------:|---------:|
> > | QServe [1] (W4A8)| 76.22|  52.13 |  77.27 |  71.36|  79.92|  72.38| 71.55|
> > | OmniQuant [2] (W4A4)|65.95|37.71| 63.26| 28.84| 72.52|  61.64| 54.99|
> > | SpinQuant [3] (W4A4)|  66.5|45.9| 75.9| 67.9| 68.0| 68.5| 65.5|
> > | ZipperQuant (W4A4)| 77.69|52.65|77.45|70.91|79.54| 72.3 | 71.76|
> >
> > *For fairness, we performed an optimal search for all smoothing factors.*
> >
> > *2 - Clarified the "W4 vs W6" concern: W4A4 GPU core + sparse CPU correction.*
> >
> > Although our evaluation includes a sparse high-bit correction branch on CPU, the GPU path is strictly W4A4: all weights stored on GPU, all GPU memory traffic, and all GPU GEMMs use 4-bit weights and 4-bit activations. The high-order bits are CPU-only, evaluated sparsely and in parallel, and do not increase GPU bit-width or memory footprint. Therefore, ZipperQuant has the same GPU operator precision and resource usage as standard W4A4 methods, which is why we classify it as a W4 method. Moreover, native 4-bit compute support is becoming increasingly available on modern hardware (e.g., INT4 Tensor Core), making W4A4 not only accurate but also a practical and forward-looking deployment choice.
> >
> > [1] QServe: W4A8KV4 Quantization and System Co-design for Efficient LLM Serving, MLsys 2025
> >
> > [2] OmniQuant: Omnidirectionally Calibrated Quantization for Large Language Models, ICLR 2024 Spotlight
> >
> > [3] SpinQuant: LLM quantization with learned rotations, ICLR 2025

---

> > > ### Author Response · Authors · 2025-11-24
> > > **Author Response to Reviewer ehPS (Part 3)**
> > >
> > > >**Question 4 - As for the system efficiency, it should have a comparison with the state-of-the-art W4A4 and W4A8 systems, rather than the relatively old system of AWQ and the full precision.**
> > >
> > > Thank you for the valuable suggestion.
> > > We additionally consider end-to-end throughput comparisons against state-of-the-art W4A4 (QuaRot [1], FlatQuant [2]), W4A8 (QServe [3]) and W4A16 (AWQ [4]) systems. The throughput results show that ZipperQuant delivers clear efficiency advantages, achieving about 7.4–13.8$\times$ higher than current W4A4 systems.
> > >
> > > | **Method** |  **bz=1** |   **bz=2**   |   **bz=4**   |   **bz=8**   |
> > > |----------:|----------:|-------------:|----------------:|--------:|
> > > | TensorRT-LLM [5] (W16A16）:|   61.18   :|   117.1   :|   228.0   :|   418.2    :|
> > > | AWQ [4] (W4A4) :|     55.2    :|    116.5    :|    191.8    :|   506.4    :|
> > > | QuaRot [1] (W4A8) :| 11.2 :| 20.4 :| 46.2 :| 101.2 :|
> > > | QServe [3] (W4A8) :| 128.9 :| 264.24 :| 520.7 :| 1015.2 :|
> > > | FlatQuant [2] (W4A4) :| 9.4 :| 17.9 :| 35.72 :| 72.66 :|
> > > | QuaRot [1] (W4A4) :| 14.9 :| 27.7 :| 55.3 :| 110.3 :|
> > > | ZipperQuant (W4A4) :|  124.8  :| 247.2  :|  407.8   :| 817.9  :|
> > >
> > > Compared with QServe, our throughput is slightly lower mainly because QServe has been heavily engineered: its inference engine is rewritten in CUDA/C++ and equipped with many operator-level kernels (e.g., fused operators, I/O–compute overlap, and optimized memory access and scheduling) to fully exploit the hardware resource. In contrast, our current ZipperQuant implementation has not yet undergone optimization at the same scale for the backend engine and kernels, which leaves a practical engineering gap. However, in principle, W4A4 requires no weight dequantization steps and less data movement than W4A8 (see the W4A4 vs. W4A8 results in QuaRot), so under comparable engineering effort, ZipperQuant is expected to have a higher theoretical speedup than QServe. Therefore, We will add more kernel- and engine-level optimizations to fully realize this headroom in the future.
> > >
> > > [1] QuaRot: Outlier-Free 4-Bit Inference in Rotated LLMs, NeurIPS 2024.
> > >
> > > [2] FlatQuant: Flatness Matters for LLM Quantization, ICML 2025.
> > >
> > > [3] QServe: W4A8KV4 Quantization and System Co-design for Efficient LLM Serving, MLsys 2025.
> > >
> > > [4] AWQ: Activation-aware Weight Quantization for LLM Compression and Acceleration, MLSys 2024.
> > >
> > > [5] TensorRT-LLM: An open-source TensorRT-based LLM inference and optimization library, NVIDIA, 2025.
> > >
> > > >**Question 5- The system overhead is not studied. What is the overhead of the H2D communication of the activation and D2H communication of the partial results? How long is the W4A4 GPU kernel and how long is the end-2-end W6A4 execution including the CPU and communication part? Without this breakdown, it is hard to believe this system is practical due to the potential overhead.**
> > >
> > > Thank you for the valuable suggestion. We added a detailed latency breakdown ($\mu$ s) on NVIDIA RTX 4090 for LLaMA3-8B, covering representative linear layers (e.g., q_proj, o_proj, k_proj, v_proj, down_proj) and batch sizes.
> > >
> > > | bz | layer | GPU INT4 GEMM | GPU Quantize | GPU Total | CPU D2H | CPU LUT | CPU H2D | CPU Total |
> > > |-----:|:------|--------------:|-------------:|----------:|--------:|--------:|--------:|----------:|
> > > | 1 | q_proj, o_proj | 116 | 78  | 194.00 | 16.0 | 165.0 | 17.0 | 198.1 |
> > > |  | k_proj, v_proj | 112 | 31  | 143.00 | 15.3 | 113.0 | 15.5 | 143.7 |
> > > |  | down_proj      | 170 | 123 | 293.00 | 22.6 | 249.0 | 23.3 | 295.0 |
> > > | 2 | q_proj, o_proj | 124 | 103 | 227.00 | 17.1 | 201.3 | 17.5 | 235.9 |
> > > |  | k_proj, v_proj | 114 | 51  | 165.00 | 15.5 | 136.7 | 15.8 | 168.1 |
> > > |  | down_proj      | 269 | 213 | 482.00 | 22.3 | 423.3 | 22.5 | 468.1 |
> > > | 4 | q_proj, o_proj | 173 | 201 | 374.00 | 19.2 | 330.0 | 19.7 | 368.9 |
> > > |  | k_proj, v_proj | 115 | 65  | 180.00 | 16.0 | 149.2 | 16.5 | 181.7 |
> > > |  | down_proj      | 452 | 312 | 764.00 | 30.6 | 522.9 | 30.2 | 583.7 |
> > > | 8 | q_proj, o_proj | 318 | 285 | 603.00 | 24.3 | 528.0 | 23.1 | 575.5 |
> > > |  | k_proj, v_proj | 127 | 93  | 220.00 | 18.1 | 196.9 | 18.8 | 233.8 |
> > > |  | down_proj      | 801 | 531 | 1332.00 | 46.2 | 1045.8 | 46.7 | 1138.6 |
> > >
> > > The results show that H2D/D2H communication is fully overlapped with GPU execution, and its raw cost accounts for only about 7%–21% of the end-to-end latency across layers and batch sizes. Importantly, the communication latency here is not bandwidth-bound; it is mainly due to kernel launch overhead, so its relative fraction naturally decreases as batch size grows. Therefore, this breakdown confirms that CPU and communication overhead do not become a bottleneck in practice, making the system practical.

---

> > > > ### Author Response · Authors · 2025-11-24
> > > > **Author Response to Reviewer ehPS (Part 4)**
> > > >
> > > > >**Question 6-The equations in section 4.1 are good! But they seem only used for the illustration that the bit-width is important. This is not a problem though.**
> > > >
> > > > We appreciate the positive feedback on Sec. 4.1. The equations there are not merely illustrative— they directly motivate our design. Theorem 4.1 decomposes the W4A4 error into four terms, from which we can precisely identify what remains after smoothing: activation-related error terms shrink, while the error becomes dominated by the magnitude and quantization error of weights. Theorem 4.2 further shows that reducing this weight error requires effectively enlarging the representational range ($q_{max}$), yet increasing bit-width directly would sacrifice GPU throughput. This gap is exactly what motivates our bit-level disaggregation: it recovers a larger effective range without raising the GPU bit-width. Finally, Theorem 4.3 establishes that this bit-plane formulation has a strictly tighter upper bound than standard W4A4 quantization, providing theoretical justification for the mechanism beyond empirical improvement.
> > > >
> > > > >**Question 7-The statement "Q_X and Q_W typically adopt the same bit-width" in L157 is not accurate, e.g., QServe uses W4A8.**
> > > >
> > > > Thank you for pointing this out. The wording at L157 was indeed ambiguous. Our intention was not to claim that $Q_X$ and $Q_W$ “typically” share the same bit-width, but to state that some prior quantization pipelines adopt equal-bit configurations, while others—such as QServe—use mixed-precision settings like W4A8. The subsequent sentence (“otherwise, one operand must be dequantized…”) already accommodates mixed-precision cases. We will revise the phrasing in the final version to remove this ambiguity and accurately reflect both equal-bit and mixed-bit designs.
> > > >
> > > > >**Question 8-It would be good to add space between the columns of Table 1. It is hard to differ different columns for its row-1**
> > > >
> > > > Thank you for the valuable suggestion. We will adjust the spacing between the columns of Table 1 to make the columns easier to distinguish.
> > > >
> > > > >**Question 9-It seems have not present the ratio of higher bits and lower bits in the evaluated models and tasks. What is the ratio of the higher bits in each portion of the weight?**
> > > >
> > > > Thank you for your valuable question.
> > > >
> > > > *1 - Explanation on higher-bit ratio: extremely sparse and tunable after smoothing*
> > > >
> > > > The higher-bit ratio corresponds to the non-zero density of the high-order bit planes $Q_{k_2}(\hat{W})$ after smoothing and bit-plane factorization. We have visualized the k_proj weight sparsity in Fig.3(b), these high-order bits are extremely sparse across layers, and this sparsity keeps both the CPU-side workload and the cross-device communication volume small. Moreover, this sparsity can be tunable via the smoothing factor, enabling an explicit accuracy–overhead trade-off.
> > > >
> > > > *2 - Additional evaluation on higher-bit sparsity*
> > > >
> > > > We further measured higher-bit sparsity on other weight matrices of Qwen3-4B. As reported in the new table, the higher-bit planes remain highly sparse for all layers, reaching over 90.32% and 99.52% sparsity on 2nd and 1st highest bits, respectively. This confirms that the observed sparsity is not specific to a single model or weight type, but a general effect of our smoothing + factorization pipeline. In the revised manuscript, we also add layer-wise higher-bit sparsity visualizations and line plots for these additional weights to make the trend clearer.
> > > >
> > > > | layer ID | bit | layer_0 | layer_3 | layer_6 | layer_9 | layer_12 | layer_15 | layer_18 | layer_21 | layer_24 | layer_27 | layer_30 | layer_33 |
> > > > |---|---|---:|---:|---:|---:|---:|---:|---:|---:|---:|---:|---:|---:|
> > > > |down_proj|1st|99.84|99.65|99.82|99.79|99.79|99.75|99.74|99.74|99.8|99.83|99.87|99.86|
> > > > |  | 2nd | 90.42 |90.6|90.49|90.52|90.48|90.6|90.6|90.55|90.57|90.54|90.43|90.41|
> > > > | gate_proj | 1st | 99.86|99.67 | 99.84 | 99.84 | 99.81 | 99.8 | 99.77 | 99.76 | 99.8 | 99.84 | 99.87 | 99.84 |
> > > > |  | 2nd | 90.34 | 92.36 | 90.37 | 90.37 | 90.39 | 90.4 | 90.45 | 90.45 | 90.39 | 90.37 | 90.35 | 90.36 |
> > > > | up_proj | 1st | 99.87 | 99.76 | 99.85 | 99.84 | 99.81 | 99.81 | 99.79 | 99.8 | 99.84 | 99.86 | 99.88 | 99.87 |
> > > > |  | 2nd | 90.32 | 90.41 | 90.35 | 90.35 | 90.41 | 90.4 | 90.43 | 90.38 | 90.36 | 90.35 | 90.32 | 90.33 |
> > > > | q_proj | 1st | 99.83 | 99.8 | 99.83 | 99.73 | 99.79 | 99.79 | 99.81 | 99.81 | 99.79 | 99.84 | 99.79 | 99.75 |
> > > > |  | 2nd | 90.35 | 90.38 | 90.4 | 90.45 | 90.39 | 90.39 | 90.38 | 90.37 | 90.39 | 90.36 | 90.42 | 90.48 |
> > > > | v_proj | 1st | 99.82 | 99.85 | 99.67 | 99.83 | 99.77 | 99.82 | 99.84 | 99.7 | 99.33 | 99.71 | 99.55 | 99.52 |
> > > > |  | 2nd |90.37|90.32|90.46|90.36|90.41|90.38|90.36|90.46|90.76|90.46|90.59|90.62|
> > > >
> > > > *For brevity, we only present the sparsity of selected layers here.
> > > >
> > > > We warmly welcome any additional comments or suggestions you may have. If our responses have addressed your concerns, could you please consider upgrading the score. Thank you very much!
> > > >
> > > > Best regards,
> > > >
> > > > The Authors

---

### Author Response · Authors · 2025-12-03
**Summary of feedback and improvements**

We sincerely thank the reviewers for their thoughtful and constructive feedback.

We are encouraged that all five reviewers recognized the **novelty** of our bit-level GPU–CPU disaggregation and LUT-based Zipper engine for low-bit LLM serving [Reviewers ehPS, xBns, PfpY, kTRi, QWVN], the **strong performance** of ZipperQuant, which consistently outperforms prior W4A4 methods and often approaches or even surpasses higher-precision baselines [Reviewers ehPS, xBns, QWVN], and the clear **systems motivation** that explicitly considers real-world hybrid CPU–GPU deployments and analyzes transfer/overlap behavior [Reviewers PfpY, kTRi, QWVN].

We greatly appreciate the reviewers' efforts in evaluating our work and the constructive suggestions are invaluable for further improving the paper, for example, adding more recent W4A4 and W4A8 baselines and the results on both workstation and server hardware [Reviewers ehPS, xBns, PfpY], streamlining the quantization formulation and notation, clarifying some minor presentation problems [Reviewers ehPS, PfpY], and refining the systems analysis with clearer latency breakdowns [Reviewers ehPS, QWVN]. These recommendations have been integrated into the revised manuscript. Below, we provide detailed point-by-point responses to each of the reviewers' comments.

---

### Meta-Review · Area_Chair_58Q5 · 2026-01-01

**Summary:**

The reviewers recognize that the paper addresses a critical challenge in post-training LLM quantization and that bit-level disaggregation between GPU and CPU can be a valuable idea. However, they identify several major issues, and some concerns were raised far more strongly and consistently than others.

The most dominant criticism concerns the lack of rigorous system-level performance analysis. Reviewers repeatedly point out that the paper does not provide a full breakdown of latency sources such as GPU kernel time, CPU LUT processing, PCIe transfer costs, and synchronization overhead. Because hybrid CPU-GPU execution is fundamentally sensitive to communication delays and limited CPU compute throughput, the absence of detailed profiling makes the reported speedups difficult to trust. This issue is intensified in decoding-heavy workloads where overlap is limited and CPU bottlenecks are more likely. This performance-analysis gap was one of the most frequently emphasized weaknesses across reviews.

Closely related to this, reviewers strongly criticize the narrow and unrealistic hardware evaluation scope. All experiments are limited to a single RTX 4090 + desktop CPU configuration. Multiple reviewers assert that the effectiveness of the method will vary substantially across other platforms, including server-grade accelerators (e.g., H100, Blackwell NVL72 setups), different CPU architectures, and systems with faster interconnects such as NVLink. They argue that without demonstrating results beyond this restricted consumer-grade setup, it is unclear whether the method is truly deployable in real LLM serving environments. This hardware-dependence concern represents another widely repeated and heavily weighted criticism.

A second cluster of concerns focuses on fairness and validity of the evaluation. Despite claiming to be a W4A4 method, the implementation uses 6-bit weight quantization, leading reviewers to question whether accuracy comparisons with true 4-bit baselines are fair. Moreover, important state-of-the-art baselines such as SpinQuant (full version), FlatQuant, QServe, TensorRT-LLM, and optimized vLLM INT4 kernels are missing, preventing proper competitive benchmarking. Many reviewers view this as a significant weakness.

Another set of comments relates to mathematical clarity and implementation details. Reviewers note inconsistencies between the error-bound theorem and its proof, ambiguous notation in the bit decomposition, and insufficient explanation of key mechanisms such as sign restoration and how the two high-order 1-bit planes are combined into a signed 2-bit representation. These gaps make it difficult to validate the technical correctness and reproducibility of the approach.

In summary, the reviewers believe the paper presents a promising direction but suffers from several critical weaknesses. Among them, the overwhelmingly most frequent and impactful concerns are the insufficient performance and latency analysis and the overly narrow hardware evaluation that does not convincingly establish real-world practicality. Strengthening these two aspects, together with fairer comparisons and clearer technical exposition, is viewed as necessary for acceptance.

**Reviewer Concerns:**

The rebuttal successfully clarified several technical points, including notational inconsistencies in the theory, details of the bit-plane implementation, the sign-restoration process, and the LUT configuration. These responses make the methodology easier to understand and reproduce.

However, the major concerns from the reviewers still remain largely unresolved. Most importantly, the system-level performance analysis is still insufficient. Although some latency numbers were added, there is still no clear profiling of CPU–GPU overlap, communication stalls, or kernel utilization — which are critical for validating any hybrid execution design.

Similarly, the hardware evaluation is still too narrow. The results are limited to specific workstation-class setups, and the authors acknowledge that their kernels are not as optimized as existing GPU-only solutions such as QServe. Because the proposed approach is highly sensitive to CPU performance and interconnect bandwidth, much broader and fairer comparisons are necessary, especially against strong W4A4 / W4A8 baselines optimized for real LLM serving environments.

Overall, while the rebuttal improves clarity, the key deployment-oriented concerns — rigorous performance breakdown and generalizability across diverse hardware — remain open and continue to limit confidence in the practical significance of the work.

**Reviewer Scores:**

Given the current state of the rebuttal and revisions, I believe it is difficult to expect that any of the reviewers would significantly change their overall score. While some minor clarifications were provided, the major concerns, particularly around system-level performance validation, fairness of evaluation, and hardware generalizability, remain largely unresolved. These issues were central to multiple reviewers’ assessments, and without substantial new evidence addressing them, meaningful score changes appear unlikely at this stage.

---

### Decision · Program_Chairs · 2026-01-26

Reject